# High-order harmonic generation in an organic molecular crystal

Falk-Erik Wiechmann [1,2], Samuel Schöpa [1], Lina Bielke [1], Svenja Rindelhardt[1,2], Serguei Patchkovskii[3], Felipe Morales [3], Maria Richter [3], Dieter Bauer [1,2] ✉ & Franziska Fennel [1,2] ✉

High-order harmonic generation (HHG) is a powerful tool for probing electronic structure and ultrafast dynamics in matter. Traditionally studied in atomic and molecular gases, HHG has recently been extended to condensed matter, enabling all-optical investigations of electronic and crystal structures. Here, we experimentally demonstrate HHG in a new class of materials: thin organic molecular crystals with perfectly aligned molecules, using pentacene as a model system. Organic molecular crystals, characterized by weak intermolecular coupling, flat electronic bands, and large unit cells, differ fundamentally from conventional covalent or ionic crystals and have attracted significant interest as promising candidates for organic electronics. We show that pentacene crystals endure laser intensities sufficient for efficient HHG up to the 17th order. The harmonic yield as a function of laser polarization reveals a strong dependence on intermolecular interactions, with higher harmonic orders particularly sensitive to both nearest- and next-nearest-neighbor couplings. Model calculations indicate that weaker intermolecular interactions necessitate probing with higher harmonic orders to resolve the crystal structure. These findings suggest that HHG may serve as a powerful tool for probing the electronic structure of organic molecular crystals, enhancing all-optical techniques for studying electronic properties and ultrafast dynamics in complex organic materials.

High-order harmonic generation (HHG) is a highly nonlinear and coherent process in which systems exposed to intense laser fields convert low-energy radiation into radiation at integer multiples of the incident photon energy. HHG enables the study of ultrafast dynamics in matter, such as laser-driven electronic processes in molecules[1–6], inorganic crystals[7–9], liquids[10,11], liquid crystals[12], transition metal dichalcogenides[13], metals[14], and topological insulators[15]. In crystalline solids, HHG provides a powerful tool for mapping neighboring atomic sites[16,17], reconstructing band structures[18], and imaging the valence electron distribution[19,20]. Therefore, HHG has been established as a powerful spectroscopic method for inorganic solid-state targets. In this work, we extend HHG to organic molecular crystals (OMCs), a

target class so far not considered in extreme nonlinear optics. Due to their low cost, light weight, and great processing versatility OMCs have attracted significant attention as promising candidates for organic electronics like thin-film field-effect transistors (OFETs)[21,22], organic light emitting diodes, sensors, and organic photovoltaics[23,24].

Given their practical significance, a detailed understanding of the charge-carrier dynamics in OMCs is highly desirable and high-harmonic spectroscopy provides a powerful all-optical approach. From a fundamental physics perspective, OMCs also represent a compelling new class of materials for HHG, as they consist of perfectly aligned organic molecules in the crystalline phase. However, their weak intermolecular coupling of only a few hundred wavenumbers

[1]Institute for Physics, University of Rostock, Rostock, Germany. [2]Department Life, Light & Matter, University of Rostock, Rostock, Germany. [3]Max Born Institute, Berlin, Germany. ✉e-mail: Dieter.bauer@uni-rostock.de; Franziska.fennel@uni-rostock.de

leads to the formation of very flat electronic bands. This poses a fundamental question: What governs the harmonic generation process in these weakly coupled systems? Is it driven primarily by the coherent response of individual molecules, or does weak intermolecular coupling influence strong-field interactions, imprinting the crystal structure onto the harmonic signatures?

Until now, OMCs have only been shown to generate perturbative, low-order harmonics[25,26]. In polycrystalline organic semiconductors, higher-order harmonics up order nine have been observed[27]. However, due to the random molecular orientation, no directional dependence on molecular states, intermolecular coupling or crystal structure was identified.

## Results

In this work, we present the generation of high harmonic orders from a single-crystalline OMC, observing harmonic orders up to 17. All harmonic orders have a distinct dependence on the polarization of the incoming laser pulse.

As a prototype OMC, we chose a pentacene crystal. The pentacene molecule consists of five linearly fused benzene rings and is a text book organic molecule due to its conjugated π-electron system. Pentacene crystals[28] attract great attention as they perform singlet fission breaking the Schockley–Queisser limit[29–31], are used in OFETs and organic light emitting diodes[32], are easy to grow[33], and are theoretically well characterized[34–36].

We consider two potential origins of the generated harmonics in OMCs: (i) the harmonics are generated independently by the well-aligned pentacene molecules. In this scenario, the measured spectrum could be understood as the coherent sum of the individual molecular contributions, and the harmonic efficiency should be enhanced for laser polarizations matching the dominant transition dipole moments of the individual molecules. (ii) The harmonic generation is strongly influenced by the crystal structure, where couplings to nearest and possibly next-nearest neighbor molecules play a decisive role. This would involve electrons sensing the crystal lattice during their dynamics, and harmonics should be enhanced for laser polarizations along directions that connect the neighbors, a behavior previously observed in inorganic crystals[16,20].

To discriminate between the two possible mechanisms, we performed polarization-dependent harmonic yield measurements from a pentacene crystal placed on a wide band-gap sapphire substrate. The driving peak field strength was 0.99 TW/cm² with a photon energy of 0.31 eV (corresponding to 4 μm wavelength). At these measurement conditions the crystal stayed intact and no modification of the crystal was observed. The photon energy of the applied field of 0.31 eV is well below the lowest electronic transition energy of 1.85 eV[34]. Odd harmonic orders from 3 to 17 were observed, collected in transmission geometry, and analyzed using grating spectrometers, see Fig. 1[37]. To investigate the process behind the harmonic generation, the harmonic yield was measured as a function of the laser polarization direction, see Fig. 2c, d, by turning the laser polarization with a λ/2-waveplate and keeping the crystal fixed. The electric field was propagating along the crystalline c-axis, see Fig. 1b, which is perpendicular to the surface of the crystal. The angle of polarization is defined with respect to the projection of the crystalline a-axis onto the plane perpendicular to the propagation direction of the pulse; for further details see "Methods".

In case the generation mechanism is dominated by isolated molecules, we expect maximum harmonic emission when the laser polarization maximally aligns with the directions of the molecular electronic transition dipole moments occurring along either the short or the long axes of the individual pentacene molecules. The first dipole-allowed electronic transition ($S_0 \rightarrow S_1$) is parallel to the short molecular axis, and the corresponding projections for the two molecules in the unit cell onto the polarization plane point at 32° and 165°, indicated by the black arrows together with dashed dotted lines in

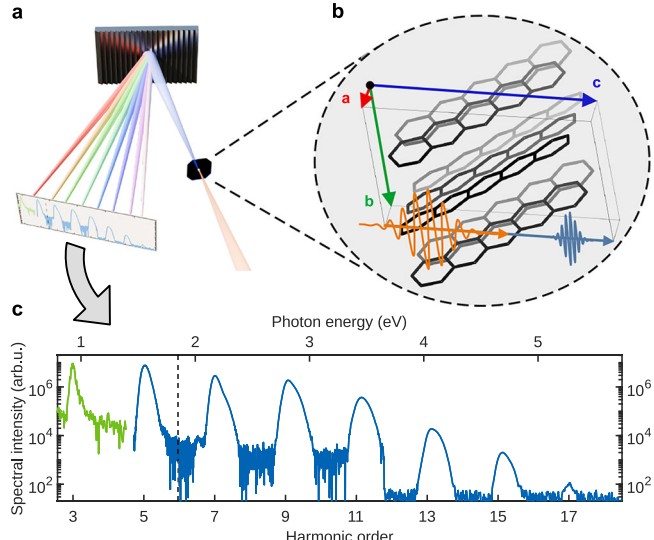

**Fig. 1 | Schematic of the experimental setup. a** A mid-IR pulse is focused onto a single pentacene crystal, inducing harmonic generation. Harmonics are collected in transmission geometry and analyzed using two diffraction grating spectrometers, covering a spectral range from 2500 to 220 nm, containing harmonic orders 2–17. **b** Crystal structure and unit cell of pentacene as obtained by X-ray diffraction. Incident and harmonic pulse are also indicated. **c** Harmonic spectrum generated by a driving field with photon energy of 0.31 eV and polarization direction at 120° relative to the **a**-axis of the crystal. The spectrum extends from the 3rd to the 17th harmonic order. Different colors represent data measured with different spectrometers. Variations in noise levels result from differing integration times.

Fig. 2a, c, d. The second dipole-allowed electronic transition ($S_0 \rightarrow S_2$) is parallel to the long molecular axis, which points in the same direction for both molecules in the unit cell. The projection of the long molecular axes on the polarization plane leads to an angle of 127°, indicated by the black arrows together with dashed lines in Fig. 2a, c, d.

On the other hand, couplings between the pentacene molecules in the crystal could enhance the harmonic emission when the polarization of the electric field aligns with the directions of the connecting lines between the nearest and next-nearest neighboring molecules, whose projections onto the plane of polarization point at 53° and 133° for the nearest and 0° for the next-nearest neighbors, indicated by the gray arrows and the dotted lines in Fig. 2b, c, d.

For all harmonic orders, we observed a pronounced dependence of the harmonic yield on the laser polarization direction (see Fig. 2c, d). The low harmonic orders 3–7 exhibit two principal emission directions, with peaks near 55° and 130° pointing in the directions of the nearest neighbors. Additionally, higher-order harmonics (orders 9–15) reveal a third emission lobe at 0°, which is the direction of the next-nearest neighbor. Interestingly, for harmonic 9 and 15 the enhancement is primarily around 55° and 130°, while the 0° lobe is strongly reduced in comparison to order 11 and 13, see Fig. 2e with logarithmic scale. The width of the emission lobes decreases with increasing harmonic order, indicating a stronger sensitivity to the crystal structure and intermolecular interactions for the high harmonic orders. The narrowing of the lobes with increasing harmonic order has also been observed for inorganic solid targets[16,20,38].

Summarizing, we find enhanced harmonic yield in the direction of nearest and next-nearest neighbors, such that we conclude that the crystal structure seems to strongly influence the generation process. Higher-order harmonics seem to be much more sensitive to the weaker next-nearest neighbor coupling compared to the lower harmonics.

In contrast, we do not find enhanced harmonic yield in the directions of the $S_0 \rightarrow S_1$ transition dipole moments pointing into 32° and 165° for the two molecules in the unit cell, which indicates that the

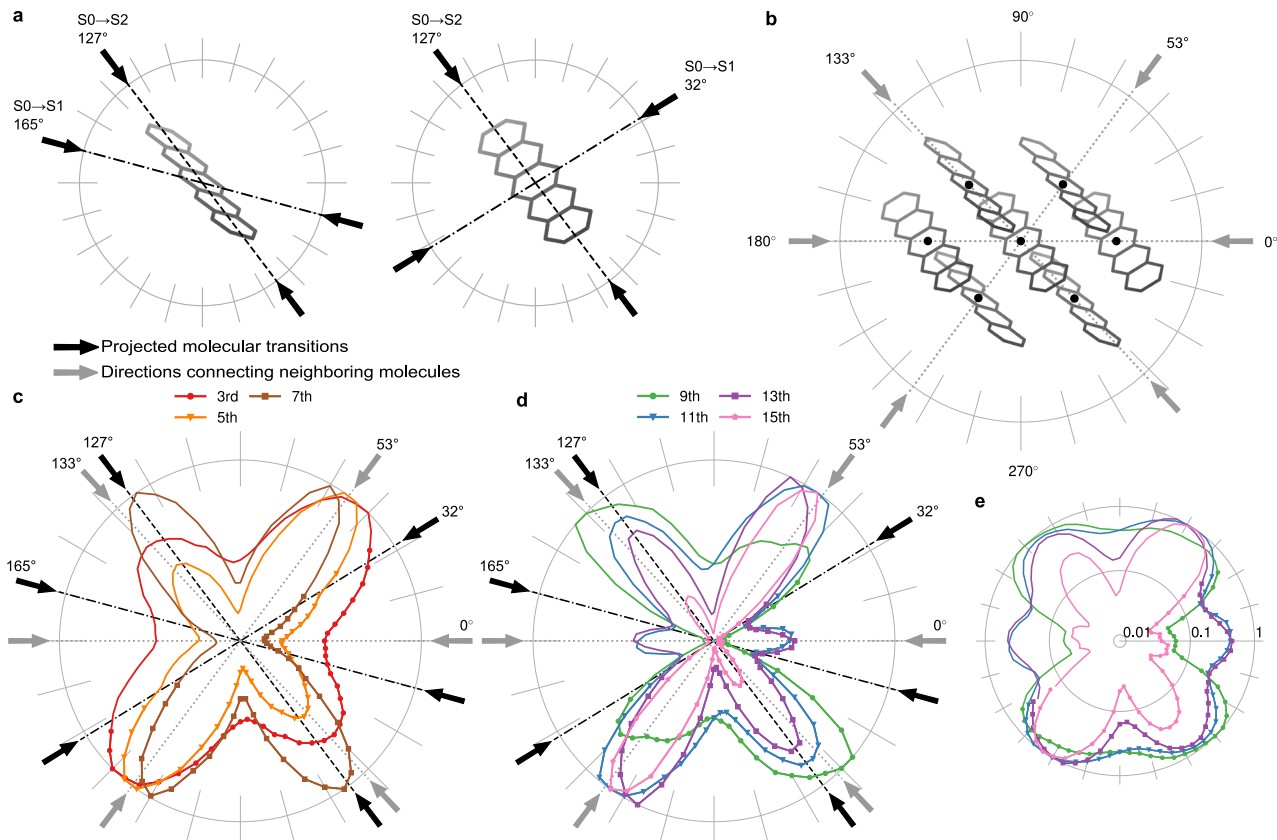

**Fig. 2 | Measured polarization-dependent harmonic yield.** In all polar plots, 0° corresponds to the projection of the crystalline **a**-axis vector onto the laser polarization plane. **a** The two different alignments of pentacene molecules, derived from the crystal structure. Black arrows represent the directions of the transition dipole moments for the single-molecule excited states, projected onto the laser polarization plane. **b** Alignment of the pentacene crystal with the laser propagating into the plane, along the crystalline **c**-axis. Gray arrows indicate the angles where the laser polarization aligns with the connecting lines between molecules. **c, d** Experimentally measured normalized harmonic yields for the 3rd– 7th harmonic and the 9th–15th harmonic, respectively. The black (single molecule) and gray (crystal structure) arrows indicate the same directions as in (**a**, **b**). **e** Polar plot like in **d** but with logarithmic scaling for a better visibility of small signals around the origin of the polar plot.

direction of this single molecule transition is not decisive for the angular dependent harmonic yield. However, we cannot rule out significant contribution of the $S_0 \to S_2$ transition at 127° experimentally, as it would be superimposed with the nearest neighbor at 133°.

To understand the origin of the measured angular dependence, we pursued three complementary theoretical approaches: first, to assess the importance of intramolecular effects of the individual molecules in the crystal for the HHG process, we have calculated high harmonic spectra generated by the combination of two noninteracting pentacene molecules with the same spatial orientations as the two differently oriented molecules in the unit cell of the crystal, applying a high-level quantum approach (multireference configuration interaction with all single excitations, MR-CIS) to calculate the electronic structure of the pentacene molecule. Second, we used TD-DFT calculations to model the response of the entire pentacene crystal, including the intermolecular effects of the coupled molecules, to discern the impact of the crystal structure on the polarization dependence of the harmonics. Third, we developed a simplified but instructive and flexible 2D tight-binding (TB) approach to systematically study the HHG mechanism by separating single-molecule effects from crystal structure effects and accounting for variable intermolecular couplings.

The dashed red curves in Fig. 3 depict the angular dependence of harmonics 3–13 generated coherently by two individual pentacene molecules as calculated with the MR-CIS approach. The two molecules had the same spatial alignment as those in the unit cell of the crystal. A 4 μm laser pulse with a peak intensity of 1 TW/cm² was used, for further

details see "Methods". For all harmonic orders, the yield is pronounced in a single dominant direction. In particular, the higher-order harmonics align closely with the long molecular axis, which corresponds to the highest polarizability, the strongest molecular response, and the transition dipole moment from the ground to the second dipole-allowed excited state $S_0 \to S_2$. Instead, for the third harmonic the $S_0 \to S_1$ transition (black arrow at 165°) seems to dominate. Crucially, the multiple-peak structure observed in the experiment is absent in the individual-molecule calculations, highlighting that the single molecule response is not sufficient to describe the experimental observations and that intermolecular couplings within the crystal must be important, despite their apparent weakness.

We tested this hypothesis using the Octopus package[39,40] to perform real-space, time-dependent density functional theory (TD-DFT) simulations of HHG in the pentacene crystal. Compared to conventional ionic- and covalently bonded solids, pentacene has a significantly larger unit cell and a greater number of occupied bands, making time-dependent HHG simulations with a long-wavelength driving pulse computationally demanding. To manage this complexity, we restricted the TD-DFT simulations to vertical transitions at the **Γ**-point.

The blue curves in Fig. 3 depict the polarization-dependent harmonic yields generated in the periodic molecular crystal. Compared to the calculated harmonic response of individual molecules, the crystal calculations reveal a much more intricate polarization dependence. Notably, harmonics 5–13 exhibit enhanced yields for three distinct

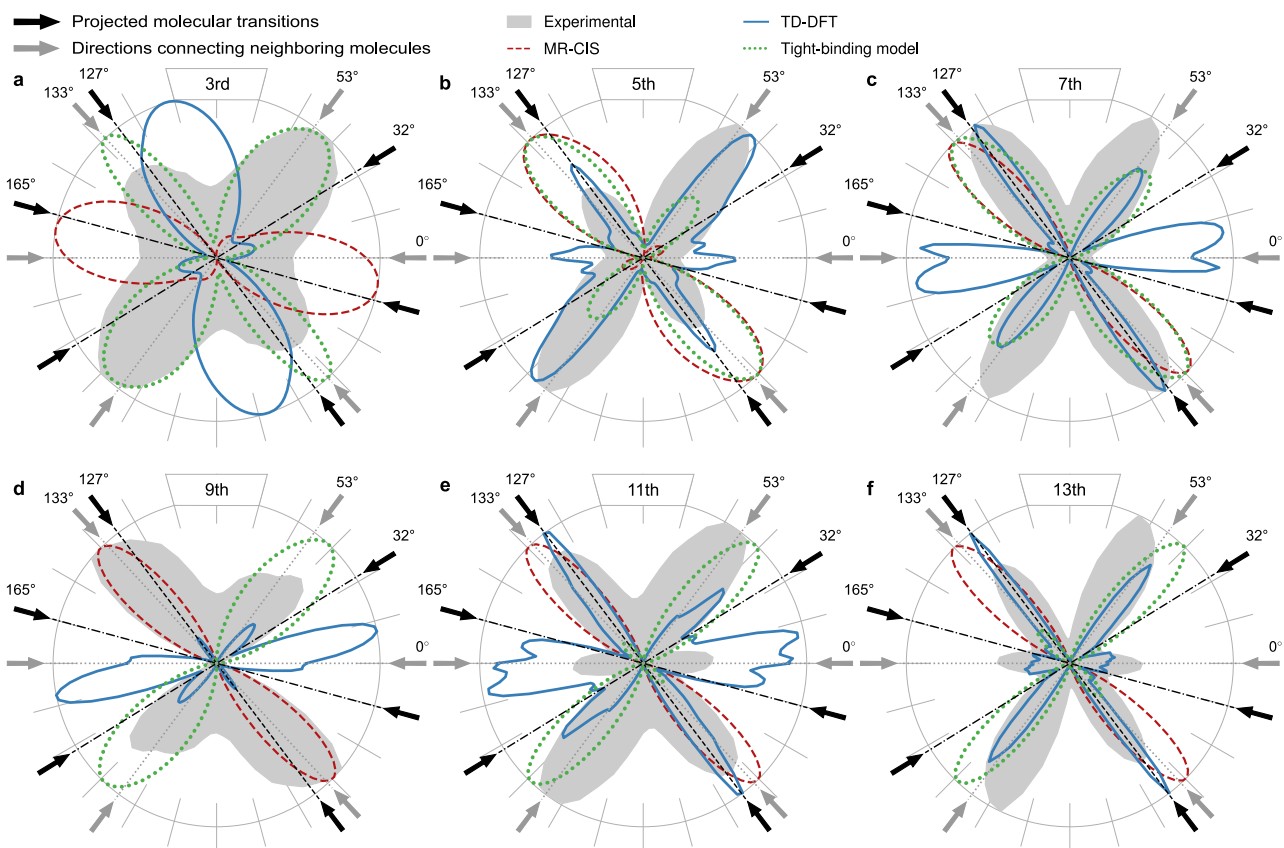

**Fig. 3 | Simulated polarization-dependent harmonic yield compared to experimental results. a–f** Polarization-dependent HHG yields of the experimental measurements for harmonic orders 3–13 (gray shaded), compared to the results of an MR-CIS simulation for the combined individual molecular response (dashed red), TD-DFT simulations (blue), and the simple 2D tight-binding model (dotted green). Gray and black arrows indicate the projected directions of the intermolecular connecting axes and the molecular transition dipole moments, respectively (see Fig. 2).

polarization directions, matching the nearest- and next-nearest-neighbor orientations, as observed experimentally.

The quantitative discrepancy between the TD-DFT calculations and experimental results is not unexpected. Specifically, the HOMO-LUMO gap of single pentacene molecules is underestimated[41]. Additionally, DFT-based methods tend to artificially redshift electronic transitions due to charge-transfer effects in herringbone dimers[42], which cannot be fully captured by local exchange-correlation functionals alone[43]. Those limitations have a particular effect on low-lying electronic states, and probably explain the deviating behavior of the third harmonic from all higher-order harmonics. Nevertheless, the qualitative polarization dependence obtained with TD-DFT for orders 5 to 13 matches the experimental observations.

The drop in harmonic efficiency between low and high orders observed for single-molecule and crystal calculations gives an additional hint to the relevant mechanism. For the individual molecule calculations, the harmonic intensity decreases about 10 orders of magnitude from harmonic 3 to 13, see Supplementary Fig. 2. Such a steep drop suggests that the higher-order harmonics would be undetectable in the experiment. In contrast, the crystal calculations exhibit a significantly smaller decrease of up to four orders of magnitude, closely matching the experimental results, see Fig. 1c. This agreement underscores the efficiency of HHG in OMCs and the crucial role of intermolecular interactions in sustaining higher-order harmonic generation in the crystal.

To gain deeper insight into the mechanism underlying the HHG process in the pentacene crystal, we constructed a simplified 2D TB model as shown in Fig. 4b. Each molecule was modeled as two TB sites, strongly coupled by the parameter $t_1$, with their molecular axes aligned

along 127°, corresponding to the 2D-projected long molecular pentacene axes onto the polarization plane of the laser field. Multiple of these molecules were placed on a regular grid such that next neighbors were located at angles of 47° and 133°, and the closest sites of neighboring toy molecules feature a weak intermolecular coupling of $t_2$. For simplicity, we omitted a coupling to next-nearest neighbors. The four sites per unit cell lead to four bands in the corresponding model crystal. As a consequence of its simplifications, the tight binding model cannot offer quantitative accuracy. The strength of the model lies in its reduced complexity and thus in its physical transparency. The model effectively isolates the key generation mechanisms while capturing the essential properties of OMCs, featuring aligned molecules with weak intermolecular couplings. The model can be generalized to other OMCs by adjusting the geometry and the intra-to-intermolecular coupling ratio.

Examining the polarization dependence predicted by this model (green dotted lines in Fig. 3), we observed distinct maxima in both intermolecular directions for all harmonic orders except the 9th. The relative peak strengths vary with harmonic order, consistent with experimental observations. The experimental peaks in 0° direction were not reproduced as we omitted the coupling in this direction. The 5th and 7th harmonic show a stronger yield for a laser polarization of about 133° because this nearest-neighbor direction coincides with the single molecule transition, which influences the lower orders. The higher orders 9–13 exhibit their maximum around the 47° direction. For equal intermolecular coupling, in this direction the distance between the coupled sites, and thus the transition dipole, is slightly larger compared to the other direction at 133°, resulting in enhanced harmonic yield.

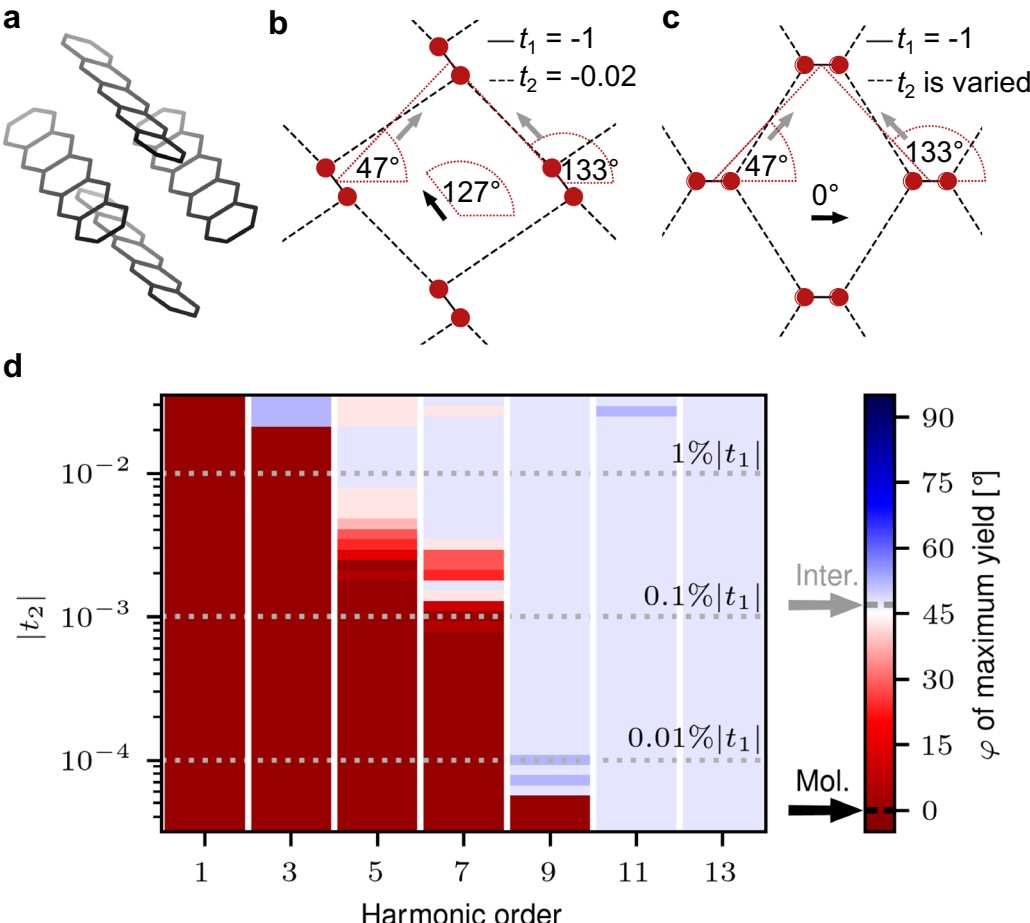

**Fig. 4 | Dependence of HHG on intermolecular coupling. a–c** Crystal structure of pentacene alongside the two variants of the simplified tight-binding model. **d** Laser polarization $\varphi$ maximizing the harmonic yield for different harmonic orders as a function of the intermolecular coupling $t_2$ in the tight-binding model shown in (**c**).

To clearly distinguish between molecular and intermolecular contributions, we rotated the molecular axes in the model away from the nearest neighbor direction to 0° as shown in Fig. 4c, ensuring that the molecular and intermolecular directions were well separated. Note that for this geometry an additional symmetry arises in which both intermolecular axes at 47° and 133° are equivalent, and it is sufficient to consider only the polarization range from 0° to 90°. We varied the intermolecular coupling $t_2$ across three orders of magnitude and calculated the polarization angle-dependent harmonic yield. Examples of the polar plots for an intermolecular coupling of $t_2 = 1\%|t_1|$ and $t_2 = 0.01\%|t_1|$ are displayed in the Supplementary Fig. 3. From these polar plots we extracted the laser polarization direction that maximizes the harmonic yield and color-coded that angle in Fig. 4d. Red color signifies a higher harmonic yield for a field polarization along the molecular direction, here 0°, while light blue signifies the intermolecular direction of 47°.

For a relatively strong intermolecular coupling, all harmonics were indeed most efficiently generated for a laser polarization along the intermolecular direction. At an intermolecular coupling strength of one percent of the molecular coupling, the fundamental and 3rd harmonic maximize along the angle of the molecular transition, while all higher orders show their maximum in the nearest-neighbor direction.

As $|t_2|$ further decreases, the maximum HHG yield flips towards the molecular direction, starting with the lower orders and gradually affecting higher orders. The details of the angular flipping not only depend on $t_2$ but also on the laser parameters because of potential (multi-photon) resonances. Such resonances lead to non-monotonous flipping as a function of $|t_2|$ for, e.g., harmonic

orders 5 and 7 in Fig. 4d. The results for the model system suggest that the intermolecular couplings in the pentacene crystal are in the regime where the harmonic yield is maximized for laser polarizations along nearest-neighbor directions and the weaker coupling in the next-nearest neighbor direction affects mainly the higher orders 11 and 13 in Fig. 3. In general, the tight binding models illustrate that HHG is extremely sensitive to the electronic intermolecular coupling.

Our experimental and theoretical results demonstrate that OMCs, with their perfectly aligned molecules on a crystal lattice, are a promising target class for efficient HHG. By applying high-harmonic spectroscopy to pentacene, we have introduced a powerful all-optical method for probing intermolecular coupling strengths. In weakly coupled systems, higher-order harmonics are required to resolve neighboring positions, while stronger coupling allows detection at lower orders. This principal behavior should be present in any OMC.

Beyond structural characterization, our approach paves the way towards time-resolved studies of ultrafast excitations in pentacene and related organic semiconducting materials. Temperature-dependent measurements could further reveal how thermal fluctuations influence intermolecular interactions, potentially uncovering phase transitions or polaronic effects.

The role of intermolecular coupling in organic electronics could be further elucidated by comparing the HHG response across polycyclic aromatic hydrocarbons or chemically modified pentacene crystals via functional group substitutions or packing variations. Such studies may inform design strategies for optimizing the HHG response in tailored semiconductors.

Extending HHG to a broader class of OMCs will advance the study of charge transport, excitonic coupling, and ultrafast carrier dynamics, reinforcing its potential as a versatile tool for all-optical spectroscopy and control in organic electronics and quantum materials.

## Methods

### Experimental harmonic-generation setup

Mid-IR pulses with a center wavelength of 4000 nm were generated by frequency conversion of the 800 nm output of a Ti: sapphire CPA system (Spitfire ACE PA, Spectra-Physics, 1 kHz). The Ti: sapphire laser output was used to pump an optical parametric amplifier (HE-Topas-Prime Plus, Light Conversion, 10 mJ) to generate the signal and idler at 1333 nm and 2000 nm, respectively. Noncollinear difference frequency generation (NDFG1K, Light Conversion) in an AGS crystal generated 4000 nm pulses with 40 µJ pulse energy. Signal, idler and the mid-IR driving pulses were separated by propagation to the harmonic generation setup. The mid-IR pulse duration (FWHM) was 103 fs, as characterized with a home built SHG autocorrelator using again an AGS crystal (AGS5002-SHG(I), Newlight Photonics Inc.). The intensity was controlled via a turnable achromatic half-waveplate (3-6 µm, B.Halle), in combination with a fixed polarizer (SIR3-5, Moxtec) placed behind. A second half-waveplate (3–6 µm, B.Halle) was used to adjust the angle between the polarization direction of the linearly polarized driving field and the targets' crystal axes. The pulses were focused by an $f$ = 5 cm plano-convex ZnSe lens (LA7656-E4, Thorlabs) to a focal spot with ~40 µm in radius (1/e²), measured with the knife-edge method. The target was positioned inside the laser focus with a nano-positioner (SLC-2460/SLC-2490, SmarAct) with sub-micrometer precision. The target positioning was monitored via a microscope with a long working distance. The emitted harmonics were collected and characterized by a grating spectrometer (Triax 190, Horiba). The dispersed harmonics were imaged onto a CCD camera (Newton DU970N-UVB, Andor Technology) with low noise level. The spectral region falling on the camera chip was adjusted by rotating the grating. The wavelength sensitivity of the optics and the camera allowed us to measure harmonics with a wavelength between 220 and 900 nm, covering the harmonic orders 17–5. For the third harmonic, with a wavelength of 1330 nm, an NIR sensitive fiber spectrometer (NIR Quest + 2.5, Ocean Optics) was used. Thanks to its large band gap[44], the sapphire substrate did not generate harmonics, even at the highest applied intensity of 0.99 TW/cm².

### Crystal growth and characterization

Pentacene crystals were grown by microspacing in-air sublimation[33]. For this purpose, pentacene powder (P0030, TCI, used without further purification) was deposited onto a microscope slide within a temperature-controlled cell (LTS-420, Linkam). A sapphire substrate (25.4 × 0.25 mm, Ted Pella) was positioned upside down ~200 µm above the microscope slide. The cell was heated with a rate of 50 °C/min from room temperature to 290 °C. Subsequently, the temperature was maintained at 290 °C for several minutes until large crystals formed on the sapphire substrate, which was monitored using a microscope. Heating was then stopped and the sample cooled down to room temperature. Cooling speed was increased by opening the temperature cell. The crystal structure was resolved by X-ray diffraction experiments conducted at room temperature to match the experimental conditions in the harmonic generation setup. A triclinic crystal structure of point group P1̄ with two pentacene molecules in the unit cell was identified, in agreement with the literature[28,45]. The obtained cell parameters, $a$ = 6.252 Å, $b$ = 7.752 Å, $c$ = 14.569 Å and $\alpha$ = 76.58°, $\beta$ = 87.52°, $\gamma$ = 84.83°, are in line with the well-known Siegrist-type pentacene polymorph[34,45].

Additionally, optical microscope pictures of the crystal were taken during the X-ray diffraction experiments for various diffraction angles. This allows us to associate the orientation of the unit cell vectors relative to the macroscopic crystal edges. The **a**- and **b**-axis lay close to the plane of the large crystal face, while the **c**-axis was parallel to the surface normal within the experimental uncertainties.

### Relation between the laser polarization plane and the crystal structure

The angle between the polarization direction of the mid-IR driving field and the pentacene molecules inside the crystal depends on the orientation of the crystal and therefore, on the unit cell vectors inside the harmonic generation setup. In our experiment, the mid-IR driving pulse was propagated along the surface normal of the crystal, i.e., parallel to the **c**-axis of the crystal. We have chosen to define the polarization direction of the field with respect to the projection of the **a**-axis of the crystal onto the polarization plane, which can be unambiguously determined experimentally: The molecular packing within the unit cell of the crystal leads to a splitting of the energetically lowest electronic transition into two Davydov components. The lower Davydov component (at ~670 nm or 1.85 eV) is polarized parallel to the **a**-axis[34].

Hence, to determine the direction of the crystal **a**-axis, a broad CaF₂ white-light continuum with known polarization orientation was used in an absorption microscope to measure the angle-dependent absorption. In Supplementary Fig. 1, the absorption spectra for different white-light polarization angles are shown for a wavelength range from 400 to 750 nm. The red spectrum exhibits maximum absorption at 670 nm, indicating a white light polarization direction parallel to the energetically lowest transition, i.e., the **a**-axis of the crystal. A microscope picture of the pentacene crystal used in the harmonic generation and absorption measurements is shown as an inset. The crystal size is ~200 × 200 × 4 µm. Holes are visible at positions where the laser intensity for driving harmonics exceeded 0.99 TW/cm² significantly, leading to local damage of the crystal.

Note that pentacene has a triclinic crystal structure and the crystal vectors **a**, **b** and **c** are not orthogonal to each other. Hence, the polarization plane of the electric field does not coincide with the (**a**, **b**)-plane of the crystal. For the analysis of the polarization dependence of the harmonics, we have used the Gram-Schmidt process to define an orthonormal coordinate system ($\hat{\boldsymbol{a}}_\mathrm{p}$, $\hat{\boldsymbol{b}}_\mathrm{p}$, $\hat{\boldsymbol{c}}_\mathrm{p}$) based on the crystal vectors **a**, **b** and **c**. Since the propagation direction of the laser pulse in the experiment coincides with the crystal **c**-axis, we have first defined the vector $\hat{\boldsymbol{c}}_\mathrm{p} = \boldsymbol{c}/\|\boldsymbol{c}\|$, where $\|*\|$ indicates the Euclidean norm. Next, we have defined the vector $\hat{\boldsymbol{a}}_\mathrm{p} = \tilde{\boldsymbol{a}}_\mathrm{p}/\|\tilde{\boldsymbol{a}}_\mathrm{p}\|$, where $\tilde{\boldsymbol{a}}_\mathrm{p} = \boldsymbol{a} - proj_{\hat{\boldsymbol{c}}_\mathrm{p}}(\boldsymbol{a})$, with $proj_{\boldsymbol{u}}(\boldsymbol{v}) = \langle\boldsymbol{v}, \boldsymbol{u}\rangle\boldsymbol{u}/\langle\boldsymbol{u}, \boldsymbol{u}\rangle$ and $\langle\boldsymbol{v}, \boldsymbol{u}\rangle$ denotes the inner product of the vectors $\boldsymbol{v}$ and $\boldsymbol{u}$. Vector $\hat{\boldsymbol{a}}_\mathrm{p}$ is the normalized projection of the **a**-axis vector onto the laser polarization plane. Finally, we have defined the vector $\hat{\boldsymbol{b}}_\mathrm{p} = \tilde{\boldsymbol{b}}_\mathrm{p}/\|\tilde{\boldsymbol{b}}_\mathrm{p}\|$, where $\tilde{\boldsymbol{b}}_\mathrm{p} = \boldsymbol{b} - proj_{\hat{\boldsymbol{c}}_\mathrm{p}}(\boldsymbol{b}) - proj_{\hat{\boldsymbol{a}}_\mathrm{p}}(\boldsymbol{b})$, which is orthogonal to $\hat{\boldsymbol{c}}_\mathrm{p}$ and $\hat{\boldsymbol{a}}_\mathrm{p}$.

The angle $\varphi$ in the polar plots in Figs. 2 and 3 corresponds to the angle between the field's polarization direction and the direction of the projection of the **a**-axis vector onto the laser polarization plane, i.e., the direction of $\hat{\boldsymbol{a}}_\mathrm{p}$. Note that as a result of our applied Gram−Schmidt process, $\varphi$ = 90° corresponds to the laser field polarized parallel to $-\hat{\boldsymbol{b}}_\mathrm{p}$, cf. Eqs. (4) and (8) below.

### HHG calculations for pairs of non-interacting pentacene molecules

First, we investigated the polarization dependence of the high-harmonic spectra generated by two non-interacting pentacene molecules with the same spatial orientations as the two molecules in the unit cell of the crystal. For that, we have employed the MR-CIS method (multireference configuration interaction with all single excitations) to calculate the electronic structure of the pentacene molecule, including the 496 lowest-energy singlet states. For all electronic structure calculations, we applied a cc-pVDZ basis set. The molecular geometry was

optimized at the MP2(fc) level and molecular orbitals were optimized using a CASSCF (complete active space self consistent field) active space with six electrons in six π-type orbitals (CAS(6,6)). The calculations were carried out in $D_{2h}$ symmetry, with the active-space orbitals having symmetries of $B_{2g}$, $A_u$, $B_{3g}$, $B_{1u}$, $B_{2g}$, and $A_u$. The state-averaged energy $\varepsilon_{S_0} + \varepsilon_{S_1}$ has been minimized with the ground state $S_0$ and the first singlet excited state $S_1$ equally weighted. The electronic states were then calculated using the MR-CIS approach, with the CASSCF as the reference and employing the GAMESS (US) quantum chemistry package[46–48]. For the MR-CIS active space, 31 and 37 active orbitals have been added below and above the CAS, respectively, giving a total of 74 active orbitals. 250 states (all spin multiplicities) were calculated per irreducible representation, giving a total of 496 energy-ordered singlet states. The calculations predict a doubly-excited dark state below the second dipole-allowed excited state. This dark state is disregarded from the state numbering.

Finally, the energy $\varepsilon_n$ of the electronic state n and the transition dipole moment $d_{nm}$ between the electronic states n and m of the pentacene molecule were determined.

To maintain phase consistency between the dipoles, they were evaluated using the package SUPERDYSON[49] developed at the MBI.

To obtain the angle-resolved high-harmonic spectra shown with dashed red lines in Fig. 3, we solved the TDSE in atomic units (au)

$$i\frac{\partial \Psi(\boldsymbol{r}, t; \varphi)}{\partial t} = [H_0 + V_1(t; \varphi)] \Psi(\boldsymbol{r}, t; \varphi) \tag{1}$$

for different laser polarization angles $\varphi$, ranging from 0° to 180°, with a step size of $\Delta\varphi = 2°$. $H_0$ denotes the field-free molecular Hamiltonian with $H_0 \Phi_n = \varepsilon_n \Phi_n$ and $V_1(t; \varphi) = -\boldsymbol{d} \cdot \boldsymbol{E}(t; \varphi)$ describes the laser-molecule interaction with the transition dipole moments

$$\boldsymbol{d}_{nm} = \int d\boldsymbol{r}\, \Phi_n(\boldsymbol{r})(-\boldsymbol{r})\Phi_m(\boldsymbol{r}). \tag{2}$$

The angle $\varphi$ is defined in the plane spanned by the vectors $\hat{\boldsymbol{a}}_p$ and $\hat{\boldsymbol{b}}_p$, which are the orthonormalized projections of the unit cell axes $\boldsymbol{a}$ and $\boldsymbol{b}$ onto the plane perpendicular to laser propagation direction, see Eq. (4) below. The vector $\hat{\boldsymbol{a}}_p$ defines our polarization angle as $\varphi = 0°$, which allows for a direct comparison to the experiment. The molecular wave function $\Psi(\boldsymbol{r}, t; \varphi)$ is expanded in the basis of the 496 electronic wave functions,

$$\Psi(\boldsymbol{r}, t; \varphi) = \sum_{n=0}^{495} c_n(t; \varphi)\, \Phi_n(\boldsymbol{r}). \tag{3}$$

The Gaussian-shaped laser pulse is given by

$$E(t; \varphi) = E_0 \left[\hat{\boldsymbol{a}}_p \cos(\varphi) - \hat{\boldsymbol{b}}_p \sin(\varphi)\right] e^{\frac{-t^2}{2\tau^2}} \cos(\omega t), \tag{4}$$

with a peak field strength of $E_0 = 0.00534$ au, corresponding to a peak intensity of $I_0 = 1$ TW/cm², and a frequency of $\omega = 0.011391$ au, corresponding to a wavelength of 4000 nm. The FWHM pulse duration was $2\sqrt{2\ln 2}\tau = 125$ fs.

The system of ordinary differential equations resulting from Eq. (1) with the ansatz (3) was solved with the explicit Runge–Kutta method of order 8(5,3)[50] on a time grid spanning 1200 fs, and with a time step of $dt = 0.11$ fs. The initial condition was $c_0 = 1$ and $c_n = 0$ for $n > 0$. The calculated energy gap $\Delta\varepsilon(S_0, S_1) = \varepsilon_{S_0} - \varepsilon_{S_1} = 3.23$ eV between states $S_0$ and $S_1$ was adjusted to the experimental value of ~2 eV by shifting the ground state energy accordingly.

The single-molecule harmonic dipole,

$$\boldsymbol{D}^m(t; \varphi) = \langle \Psi^m(t; \varphi) | \boldsymbol{d} | \Psi^m(t; \varphi) \rangle, \tag{5}$$

was calculated in the molecular frame and subsequently transformed to the laboratory frame for each of the two differently oriented molecules in the unit cell ($m = 1, 2$). The dipole has been multiplied by a mask that ramps down from 1 to 0 using a $\cos^4$-function over the last four cycles of the propagation time.

To obtain the angle-resolved total high-harmonic signal of the two non-interacting molecules $I_{N\omega}^{tot}(\varphi) = \left|\widetilde{\boldsymbol{D}}_{N\omega}^{tot}(\varphi)\right|^2$ at harmonic frequency $N\omega$ as shown in Fig. 3, we first removed the dipole components parallel to the laser propagation direction to exclude contributions not present in the experimentally observed far field. Next, we performed a Fourier transform of the sum of the laboratory-frame harmonic dipoles of the two molecules and determined the maximum value of each harmonic as a function of the polarization angle $\varphi$.

## HHG calculations for the full periodic pentacene crystal

In addition to computing the harmonic signal generated by two non-interacting pentacene molecules oriented as the molecules in the unit cell, we also calculated the harmonic response of the full pentacene crystal, incorporating intermolecular coupling.

For such a large and complex system, electronic structure and dynamics calculations at the MR-CIS or similar levels are computationally prohibitive. However, the real-space TD-DFT Octopus package[39,40] is a well-established tool for simulating the harmonic response of various crystalline targets[51,52], organic molecules[27,53], liquids[54], liquid crystals[55], and has also been applied to describe the nuclear dynamics of singlet exciton fission in pentacene crystals[31].

Our calculations included three steps: First, the experimental structure obtained by X-ray diffraction was relaxed with the Quantum Espresso package[56]. The new structure was then used for the ground state calculations in Octopus. Finally, we employed Octopus to simulate the response of the crystal to a short mid-IR laser pulse. In the last step, the electron wavefunctions were propagated by solving the time-dependent Kohn-Sham equations,

$$i\frac{\partial \psi(\boldsymbol{r}, t; \varphi)_{n, \boldsymbol{k}}}{\partial t} = \hat{H}_{KS}(\boldsymbol{r}, t; \varphi)_{\boldsymbol{k}} \psi(\boldsymbol{r}, t; \varphi)_{n, \boldsymbol{k}} \tag{6}$$

for different polarization directions $\varphi$ of the laser. Here, $\psi(\boldsymbol{r}, t; \varphi)_{n, \boldsymbol{k}}$ is a Kohn-Sham orbital in the band with index n at the point $\boldsymbol{k}$ in the first Brillouin zone. The Kohn–Sham Hamiltonian reads

$$\hat{H}_{KS}(\boldsymbol{r}, t; \varphi)_{\boldsymbol{k}} = \frac{1}{2}\left(-i\nabla + \frac{1}{c}\boldsymbol{A}(t; \varphi)\right)^2 + v_{ext}(\boldsymbol{r}, t) + v_H(\boldsymbol{r}, t)_{\boldsymbol{k}} + v_{xc}(\boldsymbol{r}, t)_{\boldsymbol{k}} \tag{7}$$

consisting of the kinetic energy term with the coupling to the vector potential $\boldsymbol{A}$, related to the electric field of the laser by $\boldsymbol{E}(t; \varphi) = -(1/c)\,\partial\boldsymbol{A}(t; \varphi)/\partial t$. Note that Octopus uses atomic units but for crystal calculations a prefactor $1/c$ is applied to the vector potential (as in cgs-units), where $c$ is the speed of light ($\simeq 137$ au). The potential $v_{ext}(\boldsymbol{r}, t)$ is determined by the fixed ions, $v_H(\boldsymbol{r}, t)_{\boldsymbol{k}}$ is the Hartree potential describing the mean-field interaction among electrons, and $v_{xc}(\boldsymbol{r}, t)_{\boldsymbol{k}}$ is the exchange-correlation potential for which we used the Perdew–Burke–Ernzerhof (PBE) functional[57]. Core electrons were included via the sg15 norm-conserving Vanderbilt pseudopotential set[58].

The long-range van der Waals interaction, which leads to crystal formation, was included via the Grimme-d3 dispersion correction with Becke-Johnson damping[59].

In the TD-DFT simulations, we applied the vector potential

$$A(t;\varphi) = \frac{cE_0}{\omega} \left[ \hat{\boldsymbol{a}}_\text{p} \cos(\varphi) - \hat{\boldsymbol{b}}_\text{p} \sin(\varphi) \right] \sin\left(\frac{\omega t}{2N_\text{p}}\right)^2 \cos(\omega t) \qquad (8)$$

with a peak field strength of $E_0 = 0.0018$ au (corresponding to a peak intensity of $I_0 = 0.11$ TW/cm$^2$) and a frequency of $\omega = 0.008046$ au (corresponding to a wavelength of $\lambda = 5663$ nm). The vectors $\hat{\boldsymbol{a}}_\text{p}$ and $\hat{\boldsymbol{b}}_\text{p}$ are defined in the previous section. The pulse duration was $N_\text{p} = 10$ cycles, corresponding to a FWHM of 95 fs. The calculated energy gap $\Delta\varepsilon(S_0, S_1) = \varepsilon_{S_0} - \varepsilon_{S_1} = 1.32$ eV for the **a**-polarized Davydov component is smaller compared to the experimental value of 1.85 eV. To maintain the same ratio between the energy gap and the driving frequency in experiment and theory, we adjusted the wavelength to $\lambda = 5663$ nm in the TD-DFT simulations. Pentacene comprises 72 atoms and 204 active electrons in the unit cell. A full time-dependent simulation describing the interaction of a system as large as pentacene with a long-wavelength laser pulse, while ensuring convergence with respect to the grid and **k**-space sampling, is computationally prohibitive. Therefore, we had to restrict ourselves to calculations for the $\Gamma$-point, which can be justified in view of the flat band structure of pentacene in the low-energy region[34]. All Octopus simulations employed a grid spacing of 0.4 au and utilized the enforced time-reversal symmetry propagator[60] with a time step of 0.1 au.

At each time step, we calculated the current and removed its component parallel to the laser propagation, as explained in the previous section. The HHG spectrum was obtained by Fourier-transforming the first derivative of the current into the frequency domain with a Hann window. Finally, the harmonic yield was determined by integrating the HHG spectrum over an interval of $1\omega$ centered around the corresponding harmonic.

## 2D tight-binding model to gain insight into the complex HHG mechanism

To gain further insight into the underlying mechanism in the generation of harmonics in OMCs, we have developed a simple but instructive 2D TB model. In this model, a single pentacene molecule is represented by two sites with a strong (intramolecular) coupling $t_1$. To model the crystal, we used a unit cell containing two of these toy molecules, with a weak (intermolecular) coupling $t_2$ between the nearest sites of neighboring molecules.

Similar to atomic units, $\hbar = |e| = 1$ is used for the TB model but the energy unit is determined by setting the intramolecular coupling $t_1$ to $-1$. The length unit results from specifying the distances between the sites (see below). Note that this model is not primarily intended for direct quantitative comparisons but rather aims to provide insight into the key features governing harmonic generation in OMCs.

The tight-binding Bloch Hamiltonian reads[61]

$$\hat{H}_{\alpha,\beta}(\mathbf{k}) = \sum_{\Delta\boldsymbol{R}} H_{\alpha,\beta}(\Delta\boldsymbol{R}) e^{i\boldsymbol{k}(\Delta\boldsymbol{R} + \boldsymbol{\tau}_\beta - \boldsymbol{\tau}_\alpha)}. \qquad (9)$$

Here, $H_{\alpha,\beta}(\Delta\boldsymbol{R})$ contains the coupling between site $\alpha$ in the unit cell at position $\boldsymbol{R}$ and site $\beta$ in the unit cell at $\boldsymbol{R}'$. The vectors $\boldsymbol{\tau}_\alpha$ and $\boldsymbol{\tau}_\beta$ specify the positions of the sites within the unit cell. The sum runs over $\Delta\boldsymbol{R} = \boldsymbol{R}' - \boldsymbol{R}$, which connects unit cells $|\Delta\boldsymbol{R}|$ apart. In our model, all vectors are 2D and we consider only neighbor-couplings, as described in main and Fig. 4. This leads to a $4 \times 4$ Bloch Hamiltonian.

The ground state was obtained by diagonalizing the Hamiltonian in Eq. (9). To calculate the response of the 2D crystal model to the laser field, the system was coupled to the laser vector potential $\boldsymbol{A}(t;\varphi)$ in the

dipole approximation by the Peierls substitution

$$\mathbf{k} \rightarrow \boldsymbol{k}(t) = \boldsymbol{k} + \boldsymbol{A}(t;\varphi) \qquad (10)$$

where $\boldsymbol{A}(t;\varphi)$ is given in Eq. (8).

The photon energy was selected so that slightly more than 6 photons fit into the band gap of the system, in accordance with the experiment. The pulse contained $N_\text{p} = 10$ cycles. The TDSE was initialized in the ground states for given **k**s with the two lower bands populated. The states were propagated over small time steps of 0.1 using the matrix exponential with the full, time-dependent Hamiltonian. The crystal momentum was sampled by a $6 \times 6$ **k**-point grid. The polarization-dependent harmonic yield was obtained from the expectation value for the current, as in the TD-DFT calculations described above.

In the model, we consider rectangular unit cells, each containing two molecules. Two different geometries within the unit cell were used, see Fig. 4b, c in main.

The polarization dependence of the harmonics calculated with the first geometry, Fig. 4b, is shown with dotted green lines in Fig. 3. Here, the unit cell was designed to mimic the unit cell structure of the pentacene crystal as closely as possible. The molecular axis was set to an angle of 127°, corresponding to the projection of the long molecular axis of the pentacene molecules in the crystal onto the 2D plane spanned by $\hat{\boldsymbol{a}}_\text{p}$ and $\hat{\boldsymbol{b}}_\text{p}$. As neither the experimental nor the theoretical results show any indication for a relevant role of the short molecular axis, and, in order to focus on the essential features, this axis was omitted in this model. The connecting lines between the centers of nearest neighbor molecules are at angles of 133° and 47°, see Fig. 4b. The slightly smaller angle of 47° compared to the experimentally determined 53° is a consequence of the rectangular shape of the unit cell. The distance between the sites belonging to the same molecule is set to a value of 2, while the centers of neighboring molecules are 9.2 length units apart (where the latter is close to the experimentally determined distances in atomic units). The ratio of these values is chosen to avoid overlapping molecules and to keep the intramolecular distance appropriately shorter than the distance between sites of different molecules in this 2D model. The intermolecular coupling was adjusted to a value of $t_2 = 0.02\, t_1$ in order to match the results as closely as possible to the experimentally found polarization dependence. For the results shown in Fig. 3, we used a field strength of $E_0 = 0.15/c$ in the given unit system, leading to clear harmonic spectra.

The close proximity of the nearest-neighbor coupling at 133° and the intramolecular $S_0$–$S_2$ coupling at 127° complicates the separation of both contributions. For a clear separation, we have constructed a second unit cell geometry, where the connecting lines between nearest neighbor molecules lie at the same angles as in the previous model system, but the molecular axes are rotated to lie along 0°, see Fig. 4c. As a result, this model system is mirror-symmetrical in $x$ and both signals of the two intermolecular directions at $\varphi = 133°$ and $\varphi = 47°$ have the same intensity.

The second model geometry allowed us to systematically study the impact of the strength of the intermolecular coupling $t_2$ on the harmonic generation, see Fig. 4d. For the $t_2$-dependent calculations, the field strength was scaled logarithmically as $E_0 = -0.04 \log_{10}(|t_2|)$ in order to obtain a sufficient harmonic yield for small $t_2$ parameters, but to avoid too strong excitations in the limited 4-band scheme for larger $t_2$ parameters. We verified that the harmonic "flipping" effect, where lower orders switch their maximum from the intermolecular to the molecular axis as the intermolecular coupling decreases, is due to the change in the coupling and not to the accompanying change in field strength.

## Data availability

The data generated in this study have been deposited in the Rosdok database under accession code https://doi.org/10.18453/rosdok_id00004817.

## Code availability

Codes used in this work can be obtained from the authors upon request.

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

## Acknowledgements

We acknowledge funding from the Deutsche Forschungsgemeinschaft via SFB 1477 "Light–Matter Interactions at Interfaces" (project no. 441234705).

## Author contributions

F.F. and D.B. developed the idea and supervised the project. F.-E.W. and F.F. performed the experiment and evaluated the data. S.S. performed the TD-DFT simulations. L.B. performed the tight-binding simulations. F.-E.W., S.R., and F.F. characterized the pentacene crystals. S.P., F.M., and M.R. contributed the single-molecule calculations. All authors discussed the results and wrote the manuscript.

## Funding

## Competing interests

The authors declare no competing interests.
