## [Transparent Peer Review File · Nature Communications]

High-Order Harmonic Generation in an Organic Molecular Crystal

Corresponding Author: Dr Franziska Fennel

Version 0:

Reviewer comments:

Reviewer #1

(Remarks to the Author)

This paper, "High-Order Harmonic generation in Organic Molecular Crystal" demonstrates that high-order harmonic generation (HHG) can be effectively applied to organic molecular crystals (OMCs), using pentacene as a model system. The authors use HHG to probe electronic structure. The study reveals that pentacene crystals can sustain laser intensities sufficient for generating harmonics up to the 17th order, with the harmonic yield strongly influenced by the nature of intermolecular coupling as shown by the polarization dependence. Based on model calculations, they infer that higher harmonic orders can be used to probe weak intermolecular coupling. The experimental methods and the computational details provided in the paper are adequate.

The detection of up to the 17th order harmonics from organic crystals is indeed interesting, and the three different computational models utilized by the authors to explain the trends are quite rigorous and capture the complex dynamics of the system qualitatively, supporting their claims.

However, the main issue with this work is the presentation. Certain parts of the explanations are unclear and require more rigorous explanation.

I recommend publication after the following revisions:

1. In Fig. 2, it is unclear what the grey arrows indicate. I am assuming the laser is rotated 360 degrees as indicated by the dots. As such, when the polarization aligns well with the crystal planes, the grey lines should align with the black lines, which is not the case here. Please clarify.
2. A brief summary of the overall findings near line 106 would be helpful. Specify the assumptions of each theory, and which harmonics align well with which. Most of it is currently in the Methods section, but the body of the paper does not convey the connections well.
3. Include the damage threshold of the material under the fluences used, and comment on the structural integrity of the crystals after measurement.
4. All figures should have font sizes increased significantly.

The title is too generic. The findings of the work are specific to pentacene, and will not necessarily extend to other organic crystals.

Reviewer #2

(Remarks to the Author)

This study provides experimental studies on HHG from organic molecular crystals and gives insights into intermolecular coupling effects and their sensitivity to higher-order harmonics. Below are comments and suggestions:

- Line 96: The claim that the 15th harmonic is highly sensitive to the crystal structure is intriguing. However, clarification is needed on why the intermolecular coupling at 0° cannot be probed by the 15th harmonic. Is there a specific reason for the absence of the 0° peak?
- Line 96: Please provide a clearer explanation of the physical significance of angular width.
- Line 98: The manuscript states that the $S_0 \rightarrow S_1$ transition has negligible impact on the HHG process. Could the authors further elaborate on this point?
- Line 111: There appears to be an error—Fig. 3 does not display the 15th harmonic. Please revise accordingly.
- Line 134: The TD-DFT calculations suggest that harmonics 5 to 13 are influenced by intermolecular coupling. However, the polarization dependence of the 3rd harmonic differs significantly. Could the authors clarify the underlying reason for this?

discrepancy?

- Line 142: A graphical comparison of the calculated HHG spectra for an individual molecule versus a crystal would strengthen the claim that higher-order harmonics serve as an effective probe for intermolecular coupling. Including such a figure would improve clarity.
- Line 163: The tight-binding model calculations predict maxima for the 5th and 7th harmonics at 133° , whereas higher orders peak at 47° . Experimental results for the 11th and 13th harmonics align well with the calculations, but the 5th and 9th harmonics exhibit noticeable deviations. Could the authors provide comments on this inconsistency?
- Line 695: In Extended Data Fig. 1, the angular values should be presented more explicitly for clarity.

Additionally, clarification is required in Fig. 4 regarding why the 11th harmonic is absent, despite t_1 being significantly larger than t_2 .

While the study contributes to the understanding of high-order harmonic generation in novel materials, quantitative inconsistencies between experimental and computational results remain. Given these points, the manuscript may be better suited for a more specialized journal (e.g., Physical Review B).

Lastly, it is recommended that a higher-resolution image be used for Extended Data Fig. 1, as this would improve the clarity of the sample representation.

Reviewer #3

(Remarks to the Author)

Version 1:

Reviewer comments:

Reviewer #1

(Remarks to the Author)

The authors have addressed the issues that I raised satisfactorily. I recommend publication without further changes.

Reviewer #2

(Remarks to the Author)

The authors have appropriately addressed the referee's questions, and I support the publication of their work in Nature Communications. However, I would like to point out one thing: the following paper discusses high harmonic generation from perovskite single crystals, focusing on the dependence of harmonic intensity on the molecular bonding direction. This is related to the current manuscript, which explores high harmonic generation dependent on intermolecular bonding. Sanari, Y. et al. Role of virtual band population for high harmonic generation in solids. Phys. Rev. B 102, 041125 (2020). It may be beneficial to consider citing this reference appropriately in the introduction or elsewhere in the manuscript.

Reviewer #3

(Remarks to the Author)

Dear Reviewers,

We are submitting a revised version of the manuscript "High-Order Harmonic Generation in Organic Molecular Crystals" by F. E. Wiechmann *et al.* Manuscript reference: NCOMMS-25-22416-T.

We are very grateful to the Reviewers for their valuable suggestions and comments that helped us to substantially improve the presentation of our work. We have followed all their recommendations, included a clearer and more rigorous explanation of our methods and findings, and amended the manuscript, figures, and the Methods section accordingly.

In the following, we present a point-by-point response to all remarks of the Reviewers and describe the corresponding changes made in the manuscript. In our reply letter, the Reviewers' comments are printed in black, our responses in blue, and explicit changes to the manuscript in red font. All modifications are also visible in a marked-up version of the manuscript.

We would also like to inform you that during the revision, we have detected a small error in the calculation of the MR-CIS-based high harmonic generation spectra for the non-interacting pentacene molecules (displayed with red dashed lines in Fig. 3). The adjustment of the calculated state energies to the experimental value, as described in lines 485-487 in the Methods section, was not carried out correctly. We have corrected the error and the new results show practically the same qualitative behavior for the polarization dependence of the high harmonic signal as the previous results. For the new results, the drop in harmonic yield between low and high orders, which we refer to in line 144 of the manuscript, has decreased from eleven to ten orders of magnitude. We have changed the text accordingly. Importantly, all our conclusions from the originally submitted manuscript remain fully unaffected by the correction. We have included the corrected results in the revised manuscript in Fig. 3 and the new Extended Data Fig. 2.

Data availability: We added a DOI link (https://doi.org/10.18453/rosdok_id00004817) for a data repository during resubmission, which will be used to make all relevant data permanently accessible. We added the statement: "The data displayed in the figures that support the findings of this study are available in Rosdok with the identifier https://doi.org/10.18453/rosdok_id00004817." As the data might still be subject to changes, we will activate the link and upload the final data package to the repository after acceptance of the article. Meanwhile, the current state is made available to you under the following link: <https://unibox.uni-rostock.de/getlink/fiMrTR4R5uUJ8P98Jv6utP/>

Code availability: We added the statement: "Codes used in this work can be obtained from the authors upon reasonable request."

We hope that the changes we have made will convince you that our manuscript in its present form is suitable for publication in Nature Communications.

Sincerely,

Franziska Fennel on behalf of the authors

Reply to Reviewer 1

This paper, "High-Order Harmonic generation in Organic Molecular Crystal" demonstrates that high-order harmonic generation (HHG) can be effectively applied to organic molecular crystals (OMCs), using pentacene as a model system. The authors use HHG to probe electronic structure. The study reveals that pentacene crystals can sustain laser intensities sufficient for generating harmonics up to the 17th order, with the harmonic yield strongly influenced by the nature of intermolecular coupling as shown by the polarization dependence. Based on model calculations, they infer that higher harmonic orders can be used to probe weak intermolecular coupling. The experimental methods and the computational details provided in the paper are adequate.

The detection of up to the 17th order harmonics from organic crystals is indeed interesting, and the three different computational models utilized by the authors to explain the trends are quite rigorous and capture the complex dynamics of the system qualitatively, supporting their claims.

However, the main issue with this work is the presentation. Certain parts of the explanations are unclear and require more rigorous explanation.

I recommend publication after the following revisions:

We thank the referee for the very positive evaluation and the valuable questions and comments that have prompted us to improve the clarity and completeness of our manuscript.

Below we respond to the Reviewer's individual comments and suggestions.

Reviewer 1 comment 1: *1. In Fig. 2, it is unclear what the gray arrows indicate. I am assuming the laser is rotated 360 degrees as indicated by the dots. As such, when the polarization aligns well with the crystal planes, the gray lines should align with the black lines, which is not the case here. Please clarify.*

We thank the Reviewer for drawing our attention to this ambiguity. We have studied the polarization dependence of the high harmonics signal as a function of the angle φ between the field's polarization direction and the projection of the crystal axis \mathbf{a} onto the plane perpendicular to the propagation direction of the laser, where the laser propagation direction coincides with the \mathbf{c} axis of the crystal, see Fig. 1b in the manuscript.

In Fig. 2 a, b, we highlight the two possibly important "players" in the analysis of the polarization angle-dependent high harmonic yield in OMCs:

- the dipole-allowed single molecule transitions in case of independent, non-interacting molecules within the crystal (Fig. 2a, black arrows) and
- the nearest and next-nearest neighbors in case of interacting molecules with weak couplings between the molecules in the crystal (Fig. 2b, grey arrows).

Considering optical transitions between electronic states of a single pentacene molecule, there are two directions for the transition dipole moments, see Fig. 2a: parallel to the short molecular axis, such as the $S_0 \rightarrow S_1$ transition, and parallel to the long molecular axis, such as the $S_0 \rightarrow S_2$ transition. The unit cell of the crystal contains two pentacene molecules whose long molecular axes are parallel to each other, but whose short molecular axes form an angle. The projections of the long molecular axes of the two molecules onto the laser's polarization plane is the same and at 127° , see the black arrows and black dashed line in Fig 2a. The projections of the short molecular axes onto the laser's polarization plane are at 165° and 32° , see black arrows and black dashed-dotted line in Fig 2a. If the harmonic response of the pentacene crystal would be dominated by the response of individual, virtually non-interacting pentacene molecules within the crystal, we would expect the

high harmonic signal to be maximized for field polarization directions pointing into the direction of the molecular transition dipoles, which are indicated by black arrows in Figs. 2 and 3.

In contrast, the gray arrows and dotted lines in Figs. 2 and 3 highlight 0° , 53° and 133° at which the connecting lines between nearest and next-nearest neighbor molecules are projected onto the laser's polarization plane, see Fig. 2b. For the connecting lines, we consider the lines between the centers of mass of the molecules.

Importantly, the projections of the directions of the short and long molecular axes of the individual molecules in the unit cell, on the one hand, and of the connecting lines of the neighboring molecules, on the other hand, *are fixed and do not align*. The angles associated with the projections of the short molecular axes (at 165° and 32°) are well separated from the angles associated with the projections of the connecting lines of the molecules (0° , 53° , and 133°). The angle associated with the projections of the long molecular axes (at 127°) is rather close to the 133° angle associated with one of the connecting lines between nearest-neighbor molecules, but also very well separated from the other angles at 53° and 0° associated with the projections of the other connecting lines between molecules. The lack of alignment between the angles marked with black and gray arrows allows us to distinguish between the effects of the intra- and of the intermolecular couplings, i.e. between the single molecule effects and the crystal structure effects, on the harmonic response of the crystal. Since the experimental lobes peak close to the gray arrows, which indicate the neighbor directions, we conclude that intermolecular couplings dominate the harmonic generation mechanism.

To clarify what the gray and black lines in Figs. 2 and 3 mean, we have made the following changes in the main text:

Page 3, end of 3rd paragraph, line 79:

"To investigate the process behind the harmonic generation, the harmonic yield was measured as a function of the laser polarization direction, see Fig. 2c, d, by turning the laser polarization with a $\lambda/2$ -waveplate and keeping the crystal fixed. The electric field was propagating along the crystalline c-axis, see Fig. 1b, which is perpendicular to the surface of the crystal. The angle of polarization is defined with respect to the projection of the crystalline a-axis onto the plane perpendicular to the propagation direction of the pulse, for further details see Methods."

Page 3, 4th and 5th paragraph, Line 82 & 87:

"In case the generation mechanism is dominated by isolated molecules, we expect maximum harmonic emission when the laser polarization maximally aligns with the directions of the molecular electronic transition dipole moments occurring along either the short or the long axes of the individual pentacene molecules. The first dipole-allowed electronic transition ($S_0 \rightarrow S_1$) is parallel to the short molecular axis, and the corresponding projections for the two molecules in the unit cell onto the polarization plane point at 32° and 165° , indicated by the black arrows together with dashed dotted lines in Fig. 2a, c, d. The second dipole-allowed electronic transition ($S_0 \rightarrow S_2$) is parallel to the long molecular axis, which point in the same direction for both molecules in the unit cell. The projection of the long molecular axes on the polarization plane leads to an angle of 127° , indicated by the black arrows together with dashed lines in Fig. 2a, c, d."

"On the other hand, couplings between the pentacene molecules in the crystal could enhance the harmonic emission when the polarization of the electric field aligns with the directions of the connecting lines between the nearest and next-nearest neighboring molecules, whose projections onto the plane of polarization point at 53° and 133° for the nearest and 0° for the next-nearest neighbors, indicated by the gray arrows and the dotted lines in Fig. 2b, c, d."

We clarify the meaning of the different colored arrows in Fig. 2c, d by adding the following sentence in the figure caption for panels 2c, d: “The black (single molecule) and gray (crystal structure) arrows indicate the same directions as in panels a and b.”

We have also labeled the black and gray arrows in Fig. 2c, d with the corresponding angles to make the figure clearer.

Moreover, for the sake of completeness, we have added the sentences “The calculations predict a doubly-excited dark state below the second dipole-allowed excited state. This dark state is disregarded from the state numbering.” in line 459 in the Methods section. We have also changed the phrases “the first electronic transition ($S_0 \rightarrow S_1$)” and “the second electronic transition ($S_0 \rightarrow S_2$)” into “the first **dipole-allowed** electronic transition ($S_0 \rightarrow S_1$)” and “the second **dipole-allowed** electronic transition ($S_0 \rightarrow S_2$)” throughout the manuscript. In addition, we have changed the text in line 119 to “the ground to the second **dipole-allowed** excited state $S_0 \rightarrow S_2$.”

Triggered by this question, we were not sure whether we had sufficiently described the details on the relation between the laser polarization and the crystal structure in the manuscript. Especially as pentacene has a triclinic crystal structure and therefore the crystal vectors \mathbf{a} , \mathbf{b} and \mathbf{c} are not orthogonal to each other. For our analysis of the angle-dependence of the harmonics, we have used the Gram-Schmidt process to define an orthonormal coordinate system ($\hat{\mathbf{a}}_p, \hat{\mathbf{b}}_p, \hat{\mathbf{c}}_p$) based on the crystal axes \mathbf{a} , \mathbf{b} and \mathbf{c} . To be precise, since the propagation direction of the laser pulse in our experiment coincides with the crystal \mathbf{c} axis, first, we have defined the vector $\hat{\mathbf{c}}_p = \mathbf{c} / \|\mathbf{c}\|$, where $\|\cdot\|$ indicates the Euclidean norm. Next, we have defined the vector $\hat{\mathbf{a}}_p$ by subtracting from vector \mathbf{a} the orthogonal projection of vector \mathbf{a} onto the line spanned by $\hat{\mathbf{c}}_p$ and normalizing the resulting vector. As a last step, we have defined vector $\hat{\mathbf{b}}_p$ by subtracting from vector \mathbf{b} both, the orthogonal projection of vector \mathbf{b} onto the line spanned by $\hat{\mathbf{c}}_p$ and the orthogonal projection of vector \mathbf{b} onto the line spanned by $\hat{\mathbf{a}}_p$, and normalizing the resulting vector. The angle in the polar plots in Figs. 2 and 3 corresponds to the angle between the field's polarization direction and the direction given by vector $\hat{\mathbf{a}}_p$, which is the projection of the \mathbf{a} -axis vector onto the laser polarization plane as indicated in the figure caption.

To make this clearer, we have added a more detailed description of the relation between the polarization plane and the crystal structure, including the definition of the angle in the polar plots depicted in Figs. 2, 3, in the Methods section. For this purpose, we have divided the section “Crystal growth and characterization” into two parts and supplemented the second part, with the title “Relation between laser polarization plane and crystal structure”, accordingly:

Crystal growth and characterization. [...] Additionally, optical microscope pictures of the crystal were taken during the X-ray diffraction experiments for various diffraction angles. This allows us to associate the orientation of the unit cell vectors relative to the macroscopic crystal edges. The \mathbf{a} - and \mathbf{b} -axis lay close to the plane of the large crystal face, while the \mathbf{c} -axis was parallel to the surface normal within the experimental uncertainties.

Relation between the laser polarization plane and the crystal structure. The angle between the polarization direction of the mid-IR driving field and the pentacene molecules inside the crystal depends on the orientation of the crystal and therefore on the unit cell vectors inside the harmonic generation setup. In our experiment, the mid-IR driving pulse was propagated along the surface normal of the crystal, i.e. parallel to the \mathbf{c} -axis of the crystal. We have chosen to define the polarization direction of the field with respect to the projection of the \mathbf{a} -axis of the crystal onto the polarization plane, which can be unambiguously determined experimentally: The molecular packing within the unit cell of the crystal leads to a splitting of the energetically lowest electronic transition into two Davydov components. The lower Davydov component (at ~ 670 nm or 1.85 eV) is polarized parallel to the \mathbf{a} -axis⁵.

Hence, to determine the direction of the crystal \mathbf{a} -axis, a broad CaF_2 white-light continuum with known polarization orientation was used in an absorption microscope to measure the angle-dependent absorption. In

Extended Data Figure 1, the absorption spectra for different white-light polarization angles (gray and red lines) are shown for a wavelength range from 400 nm to 750 nm. The red spectrum exhibits maximum absorption at 670 nm, indicating a white light polarization direction parallel to the energetically lowest transition, i.e. the **a**-axis of the crystal. A microscope picture of the pentacene crystal used in the harmonic generation and absorption measurements is shown as an inset. The crystal size is $\sim 200\mu\text{m} \times 200\mu\text{m} \times 4\mu\text{m}$. Holes are visible at positions where the laser intensity for driving harmonics exceeded 0.7 TW/cm^2 a lot, leading to local damage of the crystal.

Note that pentacene has a triclinic crystal structure and the crystal vectors **a**, **b** and **c** are not orthogonal to each other. Hence, the polarization plane of the electric field does not coincide with the (**a**, **b**)-plane of the crystal. For the analysis of the polarization dependence of the harmonics, we have used the Gram-Schmidt process to define an orthonormal coordinate system $(\hat{\mathbf{a}}_p, \hat{\mathbf{b}}_p, \hat{\mathbf{c}}_p)$ based on the crystal vectors **a**, **b** and **c**. Since the propagation direction of the laser pulse in the experiment coincides with the crystal **c**-axis, we have first defined the vector $\hat{\mathbf{c}}_p = \mathbf{c}/\|\mathbf{c}\|$, where $\|\cdot\|$ indicates the Euclidean norm. Next, we have defined the vector $\hat{\mathbf{a}}_p = \tilde{\mathbf{a}}_p/\|\tilde{\mathbf{a}}_p\|$, where $\tilde{\mathbf{a}}_p = \mathbf{a} - \text{proj}_{\hat{\mathbf{c}}_p}(\mathbf{a})$, with $\text{proj}_{\mathbf{u}}(\mathbf{v}) = \langle \mathbf{v}, \mathbf{u} \rangle \mathbf{u} / \langle \mathbf{u}, \mathbf{u} \rangle$ and $\langle \mathbf{v}, \mathbf{u} \rangle$ denotes the inner product of the vectors **v** and **u**. Vector $\hat{\mathbf{a}}_p$ is the normalized projection of the **a**-axis vector onto the laser polarization plane. Finally, we have defined the vector $\hat{\mathbf{b}}_p = \tilde{\mathbf{b}}_p/\|\tilde{\mathbf{b}}_p\|$, where $\tilde{\mathbf{b}}_p = \mathbf{b} - \text{proj}_{\hat{\mathbf{c}}_p}(\mathbf{b}) - \text{proj}_{\hat{\mathbf{a}}_p}(\mathbf{b})$, which is orthogonal to $\hat{\mathbf{c}}_p$ and $\hat{\mathbf{a}}_p$.

The angle φ in the polar plots in Figs. 2 and 3 corresponds to the angle between the field's polarization direction and the direction of the projection of the **a**-axis vector onto the laser polarization plane, i.e. the direction of $\hat{\mathbf{a}}_p$. Note that as a result of our applied Gram-Schmidt process, $\varphi = 90^\circ$ corresponds to the laser field polarized parallel to $-\hat{\mathbf{b}}_p$, cf. Eqs. (4) and (8) below."

Reviewer 1 comment 2: *A brief summary of the overall findings near line 106 would be helpful. Specify the assumptions of each theory, and which harmonics align well with which. Most of it is currently in the Methods section, but the body of the paper does not convey the connections well.*

We thank the reviewer for the valuable suggestion to provide a brief summary of our systematic study at this point of the manuscript. We have revised the addressed paragraph, which now reads:

" Summarizing, we find enhanced harmonic yield in the direction of nearest and next-nearest neighbors such that we conclude that the crystal structure seems to strongly influence the generation process. Higher-order harmonics seem to be much more sensitive to the weaker next-nearest neighbor coupling compared to the lower harmonics.

...

To understand the origin of the measured angular dependence, we pursued three complementary theoretical approaches: First, to assess the importance of intramolecular effects of the individual molecules in the crystal for the HHG process, we have calculated high harmonic spectra generated by the combination of two non-interacting pentacene molecules with the same spatial orientations as the two differently oriented molecules in the unit cell of the crystal, applying a high-level quantum approach (multireference configuration interaction with all single excitations, MR-CIS) to calculate the electronic structure of the pentacene molecule. Second, we used TD-DFT calculations to model the response of the entire pentacene crystal, including the intermolecular effects of the coupled molecules, to discern the impact of the crystal structure on the polarization dependence of the harmonics. Third, we developed a simplified but instructive and flexible 2D tight-binding approach to systematically study the HHG mechanism by separating single molecule effects from crystal structure effects and accounting for variable intermolecular couplings."

Reviewer 1 comment 3: *Include the damage threshold of the material under the fluences used, and comment on the structural integrity of the crystals after measurement.*

The reviewer points the attention to a critical point when working with organic crystals under strong excitation conditions. For pentacene photodamage occurs for high power excitation. We have not put major emphasize on measuring the damage threshold as it depends on multiple parameters like the laser pulse energy, the illumination time, the laser wavelength and the laser repetition rate. However, we established certain measures to make sure, we do not destroy the target.

The data shown in the paper were obtained by measuring the harmonic intensity as a function of the driving polarization direction. This scan was repeated at least five times, depending on the harmonic order. The results shown in the paper are averaged over all scans. We made sure that no damage of the crystal occurred by comparing all scans with each other, which show no significant decrease in harmonic yield, that would be expected in case the crystal was damaged or the pentacene molecules degraded.

Additionally, we found that a pentacene crystal starts to glow for intensities approaching values where the crystal gets destroyed. We have not unraveled the mechanism behind the emission, but we use it as a signature we want to avoid and always used intensities well below. To make sure the crystal does not glow, we used the microscope that was installed for observation of crystal positioning inside the laser beam, which we also use for permanent observation of the crystal during the measurement.

For all displayed measurements in the paper, we used a fluence of 0.67 TW/cm^2 . For this fluence the above-named criteria for an undamaged sample were fulfilled such that we are sure that we did not damage the sample.

To make clear that we did not damage the sample we added the following statement near line 75:

“The driving peak field strength was 0.67 TW/cm^2 with a photon energy of 0.31 eV (corresponding to $4 \mu\text{m}$ wavelength). **At these measurement conditions the crystal stayed intact and no modification of the crystal was observed. The photon energy of the applied field of 0.31 eV is well below the lowest electronic transition energy of 1.85 eV .**”

Notice, we have written in the Methods section “Holes are visible at positions where the laser intensity for driving harmonics exceeded 0.7 TW/cm^2 , leading to local damage of the crystal.” in the context of the image of the crystal. This might be a little misleading, as 0.7 TW/cm^2 is not the damage threshold but a lower estimate for which the crystal stays intact. Therefore, we changed the sentence to: **“Holes are visible at positions where the laser intensity for driving harmonics exceeded 0.7 TW/cm^2 significantly, leading to local damage of the crystal.”**

Reviewer 1 comment 4: *All figures should have font sizes increased significantly.*

We agree with the Reviewer and share the impression that the font sizes are too small. However, we have carefully prepared the figures in accordance with Nature Communications' style guidelines that stipulate a bold font of 8 pt for indicators and a font size of 5-7 pt for the text in the figures. In the previous version of the manuscript, we have used a font size of 7 pt for most of the text in the figures and 5-6 pt in cases where space was limited. **In the updated version of the figures, we have increased the font size of the small labels in Fig. 4 b and c.** Now all fonts in the figures have a size between 6 pt and 7 pt. If the Editor decides that further changes to the font size and type are necessary, we will of course be happy to implement them.

Reviewer 1 comment 5: *The title is too generic. The findings of the work are specific to pentacene, and will not necessarily extend to other organic crystals.*

We thank the reviewer for this comment and agree that we chose a generic title. Indeed, we have only experimentally demonstrated the generation of harmonics in pentacene. As we are very confident, though, that our experimental findings are transferable to many different OMCs, we suggest to change the title to “High-Order Harmonic Generation in an Organic Molecular Crystal”. Let us outline why we assume that our results are quite general:

We have carefully selected pentacene for our HHG study as it is a textbook example of organic molecules that forms prototypical molecular crystals. Due to the structural similarities, we expect other OMCs to respond similarly to the strong laser field.

The theoretical calculations are another strong indication that the principle behavior can be transferred to other OMCs. Whilst the TD-MR-CIS and TD-DFT calculations are specific to pentacene, our tight-binding model uses the very general approach of coupled quantum systems, which can be used to model the response of pentacene, but also of other OMCs by adapting the specific coupling regime and/or the crystal geometry. We have used the tight-binding model to investigate the effects of the relative strength between the coupling parameters t_1 of the intramolecular couplings and t_2 of the intermolecular couplings on the HHG process. Variation of t_1 and t_2 is associated with modelling different members of the class of acenes (that form stable crystal structures), where the intramolecular coupling decreases and the intermolecular coupling increases as a function of the number of linearly fused benzene rings. Using the tight-binding model, we have found that for the case of weak intermolecular coupling and lower orders, the harmonic generation process is dominated by single molecule effects, whereas for strong intermolecular coupling and high harmonic orders, the crystal structure becomes more relevant.

In addition, one could also change the geometry of the simulated system to match the crystal structure of other OMCs. The basic features will stay; harmonic generation might either be efficient for a laser polarization in the direction of the single molecule transition dipole moment for weak intermolecular coupling and low harmonic orders or for a laser polarization in the direction of the intermolecular coupling for strong intermolecular coupling and high orders.

We therefore feel confident that a general title is justified, even though, we have only demonstrated HHG for pentacene experimentally.

To emphasize the universal approach of our work, we have added the following sentences in the main text in the context of the general tight-binding model, line 158:

“The model can be generalized to other OMCs by adjusting the geometry and the intra-to-intermolecular coupling ratio.”

In the summary, line 198:

“This principal behavior should be present in any OMC.”

Reply to Reviewer 2

This study provides experimental studies on HHG from organic molecular crystals and gives insights into intermolecular coupling effects and their sensitivity to higher-order harmonics. Below are comments and suggestions:

We thank the reviewer for the careful reading of our manuscript and the valuable suggestions and comments, which prompted us to improve the clarity and completeness of the description of our results and analysis. Below we respond to the Reviewer's individual comments and suggestions.

Reviewer 2 comment 1: *Line 96: The claim that the 15th harmonic is highly sensitive to the crystal structure is intriguing. However, clarification is needed on why the intermolecular coupling at 0° cannot be probed by the 15th harmonic. Is there a specific reason for the absence of the 0° peak? See next comment.*

Reviewer 2 comment 2: *Line 96: Please provide a clearer explanation of the physical significance of angular width.*

We thank the reviewer for drawing our attention to these two issues as it addresses an important point of our paper and we like to clarify the argumentation and changes in the manuscript for these two comments together.

The 0° peak is present also for harmonic order 15, however, it is very weak and only hardly visible in the present version of Fig. 2. We modified Fig. 2 and included the polar plot from Fig. 2d on a logarithmic scale in Fig. 2e. This allows a better visibility of the small harmonic yield in the direction of 0° for harmonic order 15 and 9. In the new Fig. 2e we see a small lobe in the direction of 0° for order 15 that is app. 1/10 of the strong peak in the 53° direction.

We have rephrased the paragraph starting from line 90 to describe our experimental findings in more detail: “For all harmonic orders, we observed a pronounced dependence of the harmonic yield”

It now reads: “For all harmonic orders, we observed a pronounced dependence of the harmonic yield on the laser polarization direction (see Fig. 2c and d). The low harmonic orders 3 to 7 exhibit two principal emission directions, with peaks near 55° and 130° pointing in the directions of the nearest neighbors. Additionally, higher-order harmonics (orders 9 to 15) reveal a third emission lobe at 0°, which is the direction of the next-nearest neighbor. Interestingly, for harmonic 9 and 15 the enhancement is primarily around 55° and 130°, while the 0° lobe is strongly reduced in comparison to order 11 and 13, see Fig. 2e with logarithmic scale.”

In addition, we claimed that due to the narrower emission lobe of the 15th harmonic, the 15th indicates a stronger sensitivity to the crystal structure and intermolecular interactions. Mentioning the 15th order explicitly might be misleading in this context. Our intention was to convey that higher-orders are increasingly sensitive to the crystal structure and the width of the lobes decreases with harmonic order.

We changed the sentence from the manuscript “However, the emission lobes for harmonic 15 are noticeably narrower compared to the lower-order harmonics, indicating a stronger sensitivity to the crystal structure and intermolecular interactions.”

To “The width of the emission lobes decreases with increasing harmonic order, indicating a stronger sensitivity to the crystal structure and intermolecular interactions for the high harmonic orders.”

In the cases where the 0° lobe is visible, its strength is non-monotonous with the harmonic order, a behavior that is also present for the two other lobes in the direction of 53° and 133°. We think this is a feature of the complex dynamics that is induced by the MIR pulse in the molecular crystal. In the following, we like to explain the reasons why we expect a non-monotonous relative scaling of the different lobes with harmonic order.

First, let us consider an isolated system, such as a single and isolated pentacene molecule, and a high harmonic order well above the ionization potential. In such a regime, the strong-field approximation (e.g., the Lewenstein model) is valid, and the dynamics are dominated by the initial state and the continuum. These very high-order harmonics are largely insensitive to the finer details of the molecular structure. In contrast, lower-order harmonics, particularly those below the ionization potential, are more influenced by the discrete electronic structure, such as the HOMO–LUMO gap.

In a crystalline solid, however, there is no vacuum-like continuum. Instead, continuum states correspond to extended Bloch states in high-lying bands, which retain features of the periodic lattice. As the energy increases, these extended states become less sensitive to the details of the molecules constituting the crystal and more representative of the overall crystal structure. In addition, the dynamics in these high-lying bands are still very complex due to interferences between different pathways and multiphoton resonances. As higher harmonic orders result from transitions to higher lying bands having transition energies above the photon energy of the respective harmonic order $n\hbar\omega$, non-monotonic behavior like the dependence of the next-nearest neighbor lobe at 0° on the harmonic order is not unexpected. This can rationalize that the 15th harmonic exhibits a small lobe in the 0-degree direction, whereas for harmonic 11 and 13 it is more pronounced.

We observe a narrowing of the lobes with harmonic order in the experiment as well as in all theoretical calculations irrespective if the calculations concern the single molecule or the pentacene crystal. This has also been observed in various other experimental works on different targets. We added the following sentence and reference 36 to the manuscript: **“The narrowing of the lobes with increasing harmonic order has also been observed for inorganic solid targets.”**^{10,20,36}

From a theoretical perspective this narrowing of the lobes is also expected from perturbation theory. Imagine the wavefunction is expanded in spherical harmonics Y_m^l . As higher order harmonics involve transitions to energetically higher lying states, larger l, m are involved where the Y_m^l have narrower lobes (compare to angular distributions in photo electron emission). Alternatively, consider a real space picture, in which the electron follows an excursion path due to the acceleration by the laser field. In this picture, a low excursion distance is sufficient for low harmonic orders. A small excursion distance can be provided by a large number of trajectories with different locations of recombinations on pentacene molecules, resulting in a broad angular width around the connecting lines. In contrast, high orders require long trajectories and the details of the intermolecular potential becomes more relevant. During propagation the wave packet gets scattered and every additional scattering event will change the phase of the electronic wave packet, making it harder and harder to phase-match the emission and only a few trajectories with a small angular width remain. In the future we will use theoretical methods that work in real space to get insight into these features, however, this is beyond the scope of the paper.

Reviewer 2 comment 3: *Line 98: The manuscript states that the $S_0 \rightarrow S_1$ transition has negligible impact on the HHG process. Could the authors further elaborate on this point?*

We thank the reviewer for this suggestion and agree that our reasoning on this point needed to be improved.

The $S_0 \rightarrow S_1$ transition corresponds to the first dipole-allowed electronic transition of the single pentacene molecule. The transition dipole moment of this transition points in the direction of the short molecular axis.

Considering the two different orientations of the pentacene molecules in the crystal's unit cell, the projections of these directions onto the plane of the laser polarization correspond to the angles 32° and 165° , see the black arrows together with dashed-dotted lines in Fig. 2a.

Looking at the experimental results shown in Figs. 2c, d, and the new panel 2e, we find a negligible HH signal around those angles of the $S_0 \rightarrow S_1$ transition (32° and 165°). This was surprising to us as OMCs are characterized by their weak intermolecular coupling. Therefore, at first glance, one might expect the HH signal to be dominated by the single-molecule response of the individual molecules in the crystal rather than being sensitive to the crystal structure.

To better emphasize our finding that the polarization angle-dependent harmonic yield is a signature of the crystal structure and not of the single molecule electronic states, we have rephrased the paragraph starting at line 98 and the preceding paragraph, as already mentioned in the context of Reviewer 2 comment 1:

For all harmonic orders, we observed a pronounced dependence of the harmonic yield on the laser polarization direction (see Fig. 2c and d). The low harmonic orders 3 to 7 exhibit two principal emission directions, with peaks near 55° and 130° pointing in the directions of the nearest neighbors. Additionally, higher-order harmonics (orders 9 to 15) reveal a third emission lobe at 0° , which is the direction of the next-nearest neighbor. Interestingly, for harmonic 9 and 15 the enhancement is primarily around 55° and 130° , while the 0° lobe is strongly reduced in comparison to order 11 and 13, see Fig. 2e with logarithmic scale. The width of the emission lobes decreases with increasing harmonic order, indicating a stronger sensitivity to the crystal structure and intermolecular interactions for the high harmonic orders.

Summarizing, we find enhanced harmonic yield in the direction of nearest and next-nearest neighbors such that we conclude that the crystal structure seems to strongly influence the generation process. Higher-order harmonics seem to be much more sensitive to the weaker next-nearest neighbor coupling compared to the lower harmonics.

In contrast, we do not find enhanced harmonic yield in the directions of the $S_0 \rightarrow S_1$ transition dipole moments pointing into 32° and 165° for the two molecules in the unit cell, which indicates that the direction of this single molecule transition is not decisive for the angular dependent harmonic yield. However, we cannot rule out significant contribution of the $S_0 \rightarrow S_2$ transition at 127° experimentally, as it would be superimposed with the nearest neighbor at 133° .

At first glance, one might expect the contribution of the $S_0 \rightarrow S_1$ transition to dominate as the two molecules in the unit cell and the laser propagation direction are oriented such that the projection of the electric field onto the direction of the short molecular axis (transition dipole moment of the $S_0 \rightarrow S_1$ transition) is significantly larger than that onto the long molecular axis (transition dipole moment of the $S_0 \rightarrow S_2$ transition).

However, our single-molecule HHG calculations based on the MR-CIS description of the pentacene molecule, including the first 496 electronic states and their electric dipole couplings, show that the yields for all harmonics of order 5 and higher peak around the direction of the long molecular axis ($S_0 \rightarrow S_2$), with negligible yields around 165° and 32° , see red curves in Fig.3 of the paper. In simple terms, this finding can be understood as the result of the higher polarizability of the molecule along the long molecular axis, compensating for the smaller projection of the field onto the long molecular axis. Moreover, our TDDFT and TB model simulations confirm the negligible HH signal along the $S_0 \rightarrow S_1$ directions and demonstrate the importance of the intermolecular couplings for the HHG. To summarize, if molecular transitions were to dominate the harmonic generation process, the effects of the $S_0 \rightarrow S_2$ transition would outweigh those of the $S_0 \rightarrow S_1$ transition, which are negligible.

Reviewer 2 comment 4: *Line 111: There appears to be an error—Fig. 3 does not display the 15th harmonic. Please revise accordingly.*

We thank the Reviewer for pointing out this mistake. We have replaced the number “15” in the relevant sentence at line 111 with “13”.

Reviewer 2 comment 5: *Line 134: The TD-DFT calculations suggest that harmonics 5 to 13 are influenced by intermolecular coupling. However, the polarization dependence of the 3rd harmonic differs significantly. Could the authors clarify the underlying reason for this discrepancy?*

We thank the reviewer for raising this important point. In the TD-DFT simulation the biggest mismatch occurs for harmonic 3. The explanation for this mismatch follows our argumentation on Comments 1 and 2 of reviewer 2 on the origin of the different orders. Low orders, such as the third order, are particularly sensitive to the details of the low-lying electronic states. As we have stated in the manuscript, TD-DFT with the PBE functional results in a much smaller calculated energy gap of 1.32 eV for the a-polarized Davydov component compared to the experimental value of 1.85 eV. This is mainly attributed to the fact that local XC functionals cannot describe charge-transfer states, overestimating the molecular contributions to the first conduction band. To match the energy-gap, we have decreased the photon-wavelength in the TD-DFT simulation, which will match the resonances in the spectrum, but does not change the character of the excited states. With these limitations our TD-DFT simulations fail to describe the third harmonic. However, the qualitative polarization dependence obtained with TD-DFT, especially the importance of the directions specific to the crystal structure as opposed to those to the intramolecular transitions, is consistent with the experimental observations and provides us with valuable insights.

To explain the deviation of the third harmonic from all other higher order harmonics, we added the following sentence near line 140, and changed the following sentence slightly:

“Those limitations have a particular effect on low-lying electronic states, and probably explain the deviating behavior of the third harmonic from all higher-order harmonics. Nevertheless, the qualitative polarization dependence obtained with TD-DFT for orders 5 to 13 matches the experimental observations.”

Reviewer 2 comment 6: *Line 142: A graphical comparison of the calculated HHG spectra for an individual molecule versus a crystal would strengthen the claim that higher-order harmonics serve as an effective probe for intermolecular coupling. Including such a figure would improve clarity.*

We agree with the Reviewer that a graphical comparison would improve the clarity for the reader. We have included the figure below as Extended Data Fig. 2, which we refer to in the main text at line 145.

Extended Data Figure 2: Comparison of the experimentally measured harmonics (black) with calculated harmonics obtained for the case of pairs of non-interacting pentacene molecules based on the TD-MR-CIS single molecule calculations (red) and for the full periodic pentacene crystal based on the TD-DFT calculations (blue). All spectra are normalized to the fifth order as the orders 5 to 15 were measured with the same spectrometer. The harmonic spectra correspond to the laser polarization of 130° for all three cases displayed. Note that a direct comparison between the calculated harmonics requires caution as they are based on two fundamentally different systems with different approximations: gamma-point-only crystal TD-DFT calculations on a real-space grid with a local XC functional and single molecule quantum chemistry calculations using a localized basis. However, the comparison reveals a distinct difference between the relative scaling between the low and high orders. When comparing the drop of yield between low and high orders, the experimental and the TD-DFT crystal spectra show good agreement. In the spectrum for the pairs of non-interacting pentacene molecules, the drop between order three and order 13 is more significant, about 10 orders of magnitude. Such a strong drop is clearly not observed in experiment irrespective of a potential wavelength sensitive detection efficiency in the experiment.

Reviewer 2 comment 7: Line 163: *The tight-binding model calculations predict maxima for the 5th and 7th harmonics at 133° , whereas higher orders peak at 47° . Experimental results for the 11th and 13th harmonics align well with the calculations, but the 5th and 9th harmonics exhibit noticeable deviations. Could the authors provide comments on this inconsistency?*

We agree that the lower-order harmonics (e.g., 5th and 9th) deviate from the tight-binding model predictions, while higher orders (11th and 13th) show better agreement. We view this not as an inconsistency, but as a direct consequence of the model's simplifications.

The tight-binding model was constructed to capture the core mechanisms of high-harmonic generation using minimal parameters. A complex pentacene molecule with 36 atoms was approximated as a two-site system. Intermolecular coupling was reduced to point-like hopping between only the two closest sites of the nearest neighbors, omitting the more extended and anisotropic couplings present in the real crystal.

This reduction enables physical transparency. The model captures the connection between harmonic order and strong-field-driven electron motion, which in this case depends only on the on-site positions, coupling strengths,

and laser parameters. Our aim was not to create a tight-binding model with many free parameters, that we can fit to the experiment, but to extract qualitative insight from the simplest transferable framework.

Specifically, we fixed the intramolecular coupling to $t_1 = -1$, matched the molecular orientation and laser frequency to reproduce the experimental bandgap, and omitted next-nearest neighbor couplings. The molecular length was shortened to avoid unrealistic overlap in the 133° direction due to the model's 2D constraint. The 47° direction (which is 53° in the real crystal) was chosen to permit a rectangular unit cell. The weaker intermolecular coupling was set to $t_2 = -0.02$, and kept equal in both directions for simplicity, unlike in real pentacene crystals where direction-dependent coupling arises from different molecular orientations, positions, and orbital overlap.

This model yields the angular dependence shown in Fig. 2 in the manuscript, as well as in the extended plot shown above for clarity. As expected, the maxima for all harmonics (except the fundamental) align closely with the intermolecular directions. The fundamental peaks along the molecular axis, consistent with our rotated TB model (Fig. 4).

As the reviewer noted, the relative intensity of the two intermolecular peaks varies with harmonic order. In the model, harmonics 3 - 7 favor 133° , while harmonics 9 - 15 favor 47° . This arises from the simplified coupling scheme. In the model geometry, the 133° intermolecular distance is shorter, but both directions share the same coupling constant. As a result, hopping to the more distant site (47°) creates a larger induced dipole, and thus emits stronger harmonics at higher orders. This effect becomes more prominent at higher energy scales where longer-range electron dynamics dominate.

In the real material, this behavior is moderated. Firstly, the difference in intermolecular distances is smaller. Secondly, reduced spatial overlap in longer directions also lowers coupling strength, compensating the dipole effect. This explains why the experimental polarization dependence shows less variation in the relative strength of the maxima than the model.

In summary, the deviation between model and experiment reflects the model's minimalism. While it cannot provide quantitative accuracy, it effectively isolates key mechanisms and remains transferable to other organic molecular crystals. Its value lies in its interpretability.

We have added the following text to highlight the role of the TB model after line 158:

“As a consequence of its simplifications, the tight binding model cannot offer quantitative accuracy. The strength of the model lies in its reduced complexity and thus in its physical transparency. The model effectively isolates the key generation mechanisms, while capturing the essential properties of OMCs, featuring aligned molecules

with weak intermolecular couplings. The model can be generalized to other OMCs by adjusting the geometry and the intra-to-intermolecular coupling ratio.”

Reviewer 2 comment 8: *Line 695: In Extended Data Fig. 1, the angular values should be presented more explicitly for clarity.*

We thank the Reviewer for drawing our attention to this missing information. We have modified Extended Data Fig. 1, which now contains a figure legend specifying the different polarization angles and an improved version of the sample photo.

Reviewer 2 comment 9: *Additionally, clarification is required in Fig. 4 regarding why the 11th harmonic is absent, despite t_1 being significantly larger than t_2 .*

We are not sure what the Reviewer is referring to as harmonic 11 is present for all couplings. In Fig. 4, we color-code the angular direction of the strongest harmonic emission for a given intermolecular coupling t_2 relative to the intramolecular coupling t_1 . A bluish color indicates that the respective harmonic is maximized for a laser polarization pointing into the intermolecular direction, whereas a dark red color indicates that for the respective harmonic and intermolecular coupling the yield is maximized with a polarization pointing into the molecular direction.

In Fig. 4, we examined the influence of the intermolecular coupling onto the polarization angle-dependent harmonic yield. To do so, we have varied the strength of the intermolecular coupling t_2 in the tight-binding model by three orders of magnitude. As this plot is based on the tight-binding model sketched in Fig. 4c, the direction of the intramolecular coupling corresponds to 0° , while the direction of the nearest neighbor coupling corresponds to 47° . Due to the symmetry of the model, the response in the direction of the nearest neighbor at 133° is analog to that at 47° , so that we limited the plot to angles ranging from 0° to 90° for clarity. We find that for all values of t_2 , the maximum yield of harmonic 11 (and harmonic 13) occurs at 47° (marked by the light blue color), which corresponds to the nearest neighbor direction. This shows that the polarization dependence of the higher harmonic yield is sensitive to the crystal structure even for intermolecular couplings t_2 that are several orders of magnitude smaller than the intramolecular coupling t_1 , which is consistent with our experimental results.

To improve the clarity of the description of our analysis, we have rephrased the associated description in the main text (near line 174):

“We varied the intermolecular coupling t_2 across three orders of magnitude and calculated the polarization angle-dependent harmonic yield. Examples of the polar plots for an intermolecular coupling of $t_2 = 1\%|t_1|$ and $t_2 = 0.01\%|t_1|$ are displayed in the Extended Data Fig. 3. From these polar plots we extracted the laser polarization direction that maximizes the harmonic yield and color-coded that angle in Fig. 4d. Red color signifies a higher harmonic yield for a field polarization along the molecular direction, here 0° , while light blue signifies the intermolecular direction of 47° .”

In addition, as stated in the rephrased paragraph, we have added the raw data from which the optimal angle of harmonic generation in Fig. 4 is extracted for an intermolecular coupling of $t_2 = 1\%|t_1|$ and $t_2 = 0.01\%|t_1|$ as Extended Data Fig. 3:

Extended Data Figure 3: Polarization dependence of the harmonics obtained with the tight-binding model in Fig. 4c as a function of the intermolecular coupling parameter t_2 . **a**, Crystal structure of the tight-binding model from Fig. 4c alongside the 2D color plot from Fig. 4d showing the laser polarization φ that maximizes the harmonic yield for different harmonic orders as a function of the intermolecular coupling t_2 . The colors in the 2D color plot are extracted from the polar plots in **b**, **c** on the right as the angles at which the t_2 -dependent harmonic yields maximize, as indicated by the surrounding colorbars in **b** and **c**. **b**, **c**, Normalized yields of harmonics 1 to 13 for two selected values of t_2 , $t_2=1\%|t_1|$ (**b**) and $t_2=0.01\%|t_1|$ (**c**), as a function of laser polarization φ . The colorbar at the polar plots maps the polarization angle, for which the individual yields are maximized, to the color that is shown in **a**.

Reviewer 2 comment 10: *While the study contributes to the understanding of high-order harmonic generation in novel materials, quantitative inconsistencies between experimental and computational results remain. Given these points, the manuscript may be better suited for a more specialized journal (e.g., Physical Review B).*

Our work is the first demonstration of high-harmonic generation in an organic molecular crystal - a new, important material class for novel applications like OLEDs and field effect transistors. OMCs have inherent perfect molecular alignment and therefore allow for a detailed study of directional effects in harmonic generation without complex techniques. However, before our investigation, leading experts in the field anticipated that the low damage threshold of OMCs would limit harmonic generation to low orders, thereby providing minimal spectroscopic insight. Our findings defy these predictions, demonstrating harmonic generation up to the 17th order and uncovering rich information about intermolecular coupling. In addition, it is surprising that a weakly-bound molecular crystal with a flat band-structure nonetheless shows signatures of long-range intermolecular interactions in coherent high-order harmonics. One would have rather expected that the thermal disorder would wash out long range interactions. It is therefore important to share this novel finding promptly with a broad audience in a high-profile journal.

We have carried out high-level theoretical simulations to analyze our experimental results that clearly demonstrate the importance of the crystal structure in the high harmonic generation process - a rather surprising result in view of the weak intermolecular coupling in the crystal compared to the strong intramolecular dipoles. To further substantiate these findings and ascertain the role of intermolecular couplings, we have developed a tight-binding model that allows us to study the crystal structure effects in detail, clearly separating them from

the response of the individual, non-interacting molecules. Using this simple model, we are able to study HHG in OMCs for a wide range of parameters and develop an intuitive picture describing the underlying physics.

While there is no quantitative agreement between experiment and theory, our theoretical results provide a clear interpretation of our experimental results. The lack of quantitative agreement is not unexpected; in the manuscript, we have drawn readers' attention to this (lines 136-137) and specified the reasons (lines 126-130, 137-140, and 540-544). For a quantitative agreement a very time-consuming numerical investigation is necessary, which will be the subject of our work in the coming years. However, our theoretical work gives enough hints about the underlying physics to start formulating a plausible physical picture about what is going on.

We are convinced that our work constitutes an important and significant advance in the areas of attosecond physics, strong-field light-matter interactions, and organic optoelectronics. We demonstrated harmonic spectroscopy for complex and large molecules in solid state, which represents a crucial milestone for both controlling and probing electron dynamics in organic systems and giving new insight onto the harmonic generation mechanisms in weakly coupled systems.

Reviewer 2 comment 11: *Lastly, it is recommended that a higher-resolution image be used for Extended Data Fig. 1, as this would improve the clarity of the sample representation.*

We changed **Extended Data Fig. 1** accordingly, see also **Reviewer 2 comment 8**.

Reply to Reviewer 3

Reviewer's comment: *I co-reviewed this manuscript with one of the reviewers who provided the listed reports. This is part of the Nature Communications initiative to facilitate training in peer review and to provide appropriate recognition for Early Career Researchers who co-review manuscripts.*

We would like to thank the Reviewer for reviewing our manuscript and helping us to improve it. We appreciate this initiative by Nature Communications and are always happy to receive reviews from young scientists as, in our experience, they provide particularly valuable and detailed feedback that greatly contributes to improving the quality of the manuscript.